# The ribose methylation enzyme FTSJ1 has a conserved role in neuron morphology and learning performance

Mira Brazane[1],*, Dilyana G Dimitrova[1],*, Julien Pigeon[2], Chiara Paolantoni[3], Tao Ye[4], Virginie Marchand[5], Bruno Da Silva[1], Elise Schaefer[6], Margarita T Angelova[1], Zornitza Stark[7], Martin Delatycki[7], Tracy Dudding-Byth[8], Jozef Gecz[9], Pierre-Yves Plaçais[10], Laure Teysset[1], Thomas Préat[10], Amélie Piton[4], Bassem A Hassan[2], Jean-Yves Roignant[3,11], Yuri Motorin[12], Clément Carré[1]

FTSJ1 is a conserved human 2′-O-methyltransferase (Nm-MTase) that modifies several tRNAs at position 32 and the wobble position 34 in the anticodon loop. Its loss of function has been linked to X-linked intellectual disability (XLID), and more recently to cancers. However, the molecular mechanisms underlying these pathologies are currently unclear. Here, we report a novel FTSJ1 pathogenic variant from an X-linked intellectual disability patient. Using blood cells derived from this patient and other affected individuals carrying FTSJ1 mutations, we performed an unbiased and comprehensive RiboMethSeq analysis to map the ribose methylation on all human tRNAs and identify novel targets. In addition, we performed a transcriptome analysis in these cells and found that several genes previously associated with intellectual disability and cancers were deregulated. We also found changes in the miRNA population that suggest potential cross-regulation of some miRNAs with these key mRNA targets. Finally, we show that differentiation of FTSJ1-depleted human neural progenitor cells into neurons displays long and thin spine neurites compared with control cells. These defects are also observed in Drosophila and are associated with long-term memory deficits. Altogether, our study adds insight into FTSJ1 pathologies in humans and flies by the identification of novel FTSJ1 targets and the defect in neuron morphology.

## Introduction

RNA modifications represent a novel layer of post-transcriptional gene regulation (Saletore et al, 2012; Angelova et al, 2018; Zhao et al, 2020). Because of their variety and dynamic nature, they rapidly adapt gene expression programs in response to developmental changes or environmental variations. One of the most abundant RNA modifications is 2′-O-methylation (ribose methylation, Nm). Nm can affect the properties of RNA molecules in multiple ways, for example, stability, interactions, and functions (Kawai et al, 1992; Kurth & Mochizuki, 2009; Lacoux et al, 2012). Nm residues are abundant in ribosomal RNAs and tRNAs (Erales et al, 2017; Marchand et al, 2017), but are also found in other RNA types such as small nuclear RNAs (Darzacq, 2002; Dai et al, 2017), small non-coding RNAs (Li et al, 2005; Yu et al, 2005; Horwich et al, 2007; Saito et al, 2007; Kurth & Mochizuki, 2009), and mRNAs (Darzacq, 2002; Dai et al, 2017; Bartoli et al, 2018 Preprint). Many Nm positions are conserved through evolution, and their presence is essential for maintaining healthy physiological functions. Eukaryotic mRNAs are 5′-end–capped with a 7-methylguanosine ($m^7G$), which is important for processing and translation of mRNAs. In addition, Cap methyltransferases catalyse Nm of the first and second transcribed nucleotides and were shown to be important for innate immune surveillance, and neuronal development and activity (Lee et al, 2020; Haussmann et al, 2022). The loss of certain Nm modifications and/or Nm-modifying enzymes has been associated with various pathological conditions (reviewed in Dimitrova et al [2019]),

[1]Transgenerational Epigenetics & Small RNA Biology, Sorbonne Université, Centre National de la Recherche Scientifique, Laboratoire de Biologie du Développement - Institut de Biologie Paris Seine, Paris, France    [2]Paris Brain Institute-Institut du Cerveau (ICM), Sorbonne Université, Inserm, CNRS, Hôpital Pitié-Salpêtrière, Paris, France    [3]Center for Integrative Genomics, Génopode Building, Faculty of Biology and Medicine, University of Lausanne, Lausanne, Switzerland    [4]Institute of Genetics and Molecular and Cellular Biology, Strasbourg University, CNRS UMR7104, INSERM U1258, Illkirch, France    [5]Université de Lorraine, CNRS, INSERM, EpiRNASeq Core Facility, UMS2008/US40 IBSLor,Nancy, France    [6]Service de Génétique Médicale, Hôpitaux Universitaires de Strasbourg, Institut de Génétique Médicale d'Alsace, Strasbourg, France    [7]Victorian Clinical Genetics Services, Murdoch Children's Research Institute, Melbourne, Australia; Department of Paediatrics, The University of Melbourne, Melbourne, Australia    [8]University of Newcastle, Newcastle, Australia    [9]Adelaide Medical School and Robinson Research Institute, The University of Adelaide; South Australian Health and Medical Research Institute, Adelaide, Australia    [10]Energy & Memory, Brain Plasticity Unit, CNRS, ESPCI Paris, PSL Research University, Paris, France    [11]Institute of Pharmaceutical and Biomedical Sciences, Johannes Gutenberg-University Mainz, Mainz, Germany    [12]Université de Lorraine, CNRS, UMR7365 IMoPA, Nancy, France

Correspondence: clement.carre@gmail.com; clement.carre@sorbonne-universite.fr
*Mira Brazane and Dilyana G Dimitrova contributed equally to this work

including cancers (Liu et al, 2017; El Hassouni et al, 2019; He et al, 2020; Marcel et al, 2020) and brain diseases (Jia et al, 2012; Abe et al, 2014; Guy et al, 2015; Cavaillé, 2017).

FTSJ1 is a human tRNA 2′-O-methyltransferase (Nm-MTase), which belongs to the large phylogenetically conserved super-family of RrmJ/fibrillarin RNA methyltransferases (Bügl et al, 2000; Feder et al, 2003). Human male individuals bearing a hemizygous loss of function variant in the *FTSJ1* gene suffer from significant limitations both in intellectual functioning and in adaptive be-haviour (Freude et al, 2004; Froyen et al, 2007; Guy et al, 2015). Similar phenotypes, including impaired learning and memory capacity, were recently observed in *Ftsj1* KO mice that also present a reduced body weight and bone mass, and altered energy metabolism (Jensen et al, 2019; Nagayoshi et al, 2021). In flies, we recently showed that the loss of the two *FTSJ1* homologs (i.e., *Trm7_32* and *Trm7_34*) provokes reduced lifespan and body weight and affects RNAi antiviral defences and locomotion (Angelova et al, 2020). Finally, *Ftsj1* mutants in yeast (Δ*trm7*) grow poorly because of a constitutive general amino acid control activation and the possible reduced availability of aminoacylated tRNA$^{Phe}$ (Pintard et al, 2002; Guy et al, 2012; Han et al, 2018). Interestingly, this growth phenotype can be rescued by human FTSJ1, indicating a conserved evolu-tionary function.

Most of the knowledge on FTSJ1's molecular functions is derived from yeast studies. Trm7 in *Saccharomyces cerevisiae* methylates positions 32 and 34 in the anticodon loop (ACL) of specific tRNA targets: tRNA$^{Phe(GAA)}$, tRNA$^{Trp(CCA)}$, and tRNA$^{Leu(UAA)}$ (Pintard et al, 2002; Guy et al, 2012). To achieve 2′-O-methylation, Trm7 teams up with two other proteins: Trm732 for the methylation of cytosine at position 32, and Trm734 for the methylation of cytosine or guanine at position 34 (Guy et al, 2012; Li et al, 2020). The presence of both Cm$_{32}$ and Gm$_{34}$ in tRNA$^{Phe(GAA)}$ is required for efficient conversion of m$^1$G$_{37}$ to wybutosine (yW$_{37}$) by other proteins. This molecular cir-cuitry is conserved in the phylogenetically distinct *Schizo-saccharomyces pombe* and humans (Noma et al, 2006; Guy et al, 2015; Guy & Phizicky, 2015; Li et al, 2020). In *Drosophila*, we found that Trm7_32 and Trm7_34 modify, respectively, positions 32 and 34 in the ACL on tRNA$^{Phe(GAA)}$, tRNA$^{Trp(CCA)}$, and tRNA$^{Leu(CAA)}$ (Angelova et al, 2020). In this organism, we also identified novel tRNA tar-gets for these two enzymes (tRNA$^{Gln(CUG)}$ and tRNA$^{Glu(CUC)}$), which raised the question about their conservation in humans. A recent publication reported that human FTSJ1 modifies position 32 of another tRNA$^{Gln}$ isoacceptor, tRNA$^{Gln(UUG)}$ (Li et al, 2020). This study performed in HEK293T cells tested a selected subset of tRNAs using tRNA purification followed by MS analysis. It was shown that position 32 of tRNA$^{Arg(UCG)}$, tRNA$^{Arg(CCG)}$, and tRNA$^{Arg(ACG)}$, and position 34 of tRNA$^{Arg(CCG)}$ and tRNA$^{Leu(CAG)}$ are also 2′-O-methylated by human FTSJ1. tRNA$^{Arg(ACG)}$ was originally identified as a target of fly Trm7_32 (Angelova et al, 2020), whereas human tRNA$^{Leu(CAA)}$ (Kawarada et al, 2017) and yeast tRNA$^{Leu(UAA)}$ (Guy et al, 2012) were predicted targets of FTSJ1 and Trm7, respectively. However, a comprehensive and unbi-ased (not selected) analysis of all possible FTSJ1 tRNA targets was not performed, particularly in human patient samples, leaving the full spectrum of FTSJ1 tRNA substrates yet to be identified.

Previously, the enzymatic activity of mammalian FTSJ1 on se-lected tRNAs has been revealed through HPLC (Guy et al, 2015) and more recently through ultra-performance liquid chromatography–

mass spectrometry/mass spectrometry (Li et al, 2020; Nagayoshi et al, 2021). Both approaches analyse mononucleotides derived from selected tRNAs and are based on already reported sequences. The exact position of the modified nucleotide was thus inferred from available information on tRNA sequences and modification profiles database (Jühling et al, 2009; Chan & Lowe, 2016; Boccaletto et al, 2018). Recently, a new method called Ribo-MethSeq was established and allows the identification of Nm sites in a complete unbiased manner, based on the protection conferred by the ribose methylation to alkaline digestion (Marchand et al, 2016, 2017). This offers the possibility to identify every Nm site regulated by a particular enzyme, especially when investigating abundant RNAs, such as tRNAs.

In this study, we took advantage of this novel approach to identify the full set of FTSJ1's tRNA targets in humans. We report a novel *FTSJ1* pathogenic variant from an X-linked intellectual dis-ability (XLID) patient. Using blood cells derived from this affected individual and other individuals carrying distinct *FTSJ1* mutations, we performed an unbiased and comprehensive RiboMethSeq analysis to map the ribose methylation on all tRNAs and revealed new targets. In addition, we performed a transcriptome analysis in these FTSJ1-depleted cells and found that several genes previ-ously associated with intellectual disability (ID) and cancers were deregulated. We also found changes in the miRNA population that suggest potential cross-regulation of some miRNAs with these key mRNA targets. Finally, in accordance with the known importance of FTSJ1 during brain development in mice and its involvement in ID in humans, we showed that human neural progenitor cells (NPCs) with inactivated FTSJ1 present abnormal neurite morphology. We also observed this phenotype in *Drosophila* and a specific deficit in long-term memory. Altogether, our study reveals new targets potentially involved in FTSJ1 pathologies in humans and demonstrates a con-served function in neuron morphology and function.

## Results

### Comprehensive identification of human FTSJ1 tRNA targets

To identify new tRNA targets of human FTSJ1, we compared the Nm modification profiles of positions 32 and 34 for all detectable tRNA species in human lymphoblastoid cell lines (LCLs) obtained from control individuals (n = 4) versus LCLs obtained from individuals with XLID harbouring loss of function and pathogenic variants in *FTSJ1* (n = 5, from four unrelated families) (Table 1). Four of these affected individuals were already described and harbour distinct molecular defects: a splice variant leading to a premature stop codon (Freude et al, 2004) (LCL65AW and LCL65JW), a deletion encompassing *FTSJ1* and its flanking gene *SLC38A5* (Froyen et al, 2007) (LCL11), and a missense variant (p.Ala26Pro) affecting an amino acid located close to FTSJ1 catalytic pocket, resulting in the loss of Gm$_{34}$, but not of Cm$_{32}$, in human tRNA$^{Phe}$ (Guy et al, 2015) (LCL22). The last individual was not reported nor characterized before. This patient presents mild ID and behavioural manifesta-tions and harbours a de novo pathogenic variant affecting the consensus acceptor splice site of exon 6 (NM_012280.3: c.362-2A > T) (LCL-MM). This mutation leads to the skipping of exon 6 in the mRNA

**Table 1. FTSJ1 targets tRNA[Phe] at positions 32 and 34 in humans.**

| Individual | Cm$_{32}$ | Gm$_{34}$ | LCL code name |
|---|---|---|---|
| Control individuals | Present | Present | LCL16, LCL18, LCL24, LCL54 |
| Affected individuals with *FTSJ1* variant | Absent | Absent | LCL65AW, LCL65JW, LCL11, LCLMM |
| Affected individual with *FTSJ1* variant | Present | Absent | LCL22 |

Control and affected *FTSJ1* individuals' Nm status at positions 32 and 34 of human tRNA[Phe].

(r.362_414del) leading to a frameshift and a premature stop codon (p.Val121Glyfs*51) (Fig S1A). *FTSJ1* mRNA steady-state level in LCL-MM was significantly reduced when compared to LCL from control individuals (Fig S1B). In addition, treating the LCL-MM cells with cycloheximide to block translation, and thus the nonsense-mediated mRNA decay (NMD) pathway (Tarpey et al, 2007), led to an increase in *FTSJ1* mRNA abundance (Fig S1C). This result suggests that *FTSJ1* mRNA from LCL-MM cells is likely degraded via the NMD pathway.

To obtain a comprehensive picture of the Nm-MTase specificity for FTSJ1 in vivo, we performed RiboMethSeq analysis on LCLs isolated from affected individuals described above and compared with LCLs from healthy individuals. RiboMethSeq allows tRNA-wide Nm detection based on random RNA fragmentation by alkaline hydrolysis followed by library preparation and sequencing ([Marchand et al, 2017] and see the Materials and Methods section). Using this approach, we could confirm the known FTSJ1 targets (e.g., tRNA[Phe(GAA)] and tRNA[Trp(CCA)]) and assign the FTSJ1-deposited Nm modifications to their predicted positions in the ACL (C$_{32}$ and N$_{34}$; Fig 1). However, using only the MethScore calculation we could not detect a variation for Cm$_{32}$ in tRNA[Phe(GAA)]. This scoring strategy shows its limits in some particular situations as MethScore is calculated for two neighbouring nucleotides; thus, the simultaneous loss of two closely located Nm residues (e.g., Cm$_{32}$ and Gm$_{34}$ in tRNA[Phe]) makes the analysis of MethScore misleading (Angelova et al, 2020). Moreover, the presence of multiple RT-arresting hypermodifications (e.g., m$^1$G37/o2yW37 [Anreiter et al, 2021]) in the same

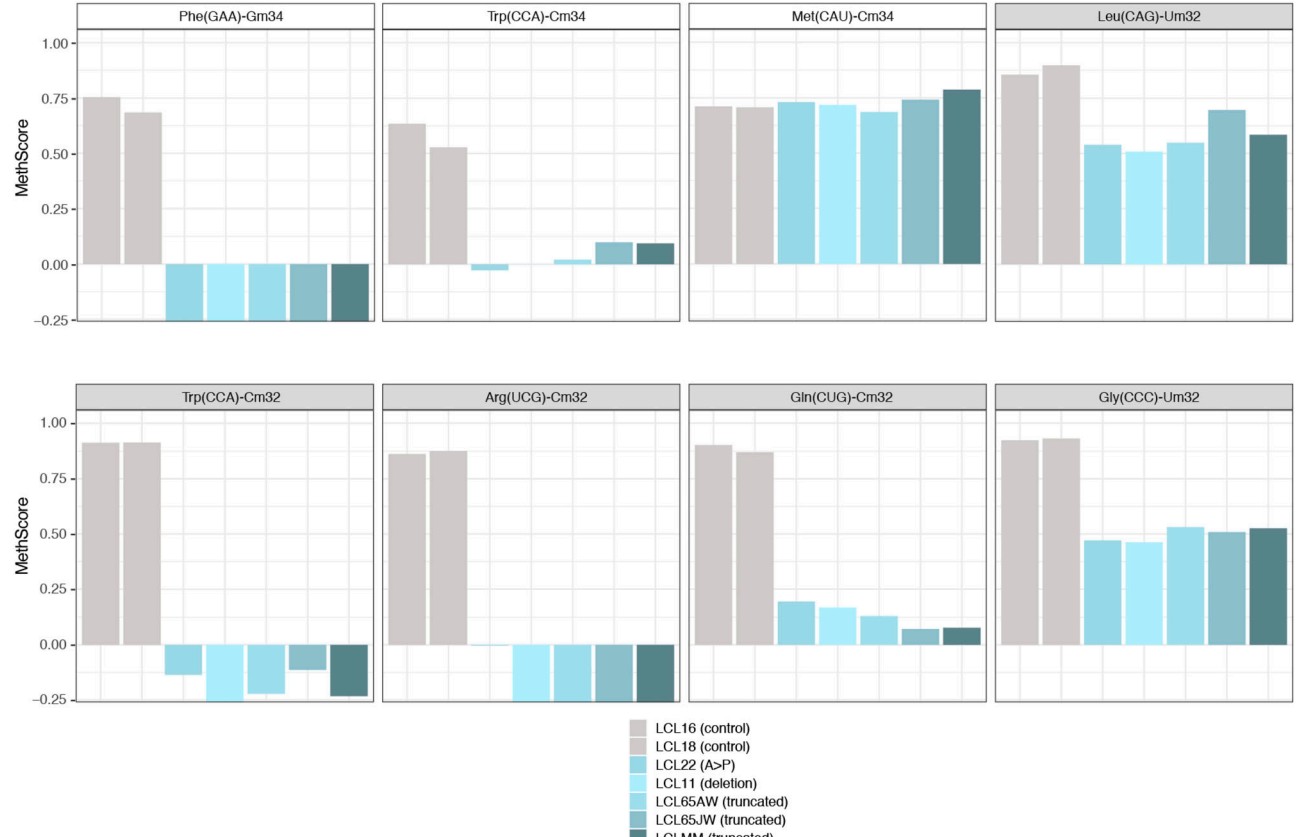

**Figure 1. FTSJ1 targets multiple tRNAs at positions 32 and 34 in humans.**
Methylation scores (MethScore) for 2'-O-methylated positions in tRNAs showing altered methylation in *FTSJ1* loss of function mutant LCLs. MethScore (Score C), representing the level of ribose methylation, was calculated from protection profiles. Data are shown for positions 32 and 34 in different *H. sapiens* tRNAs as measured in different LCL lines that are indicated with a different colour code. Grey: control LCL; blue: *FTSJ1* mutant LCLs. Met(CAU)-Cm34 is not deposited by FTSJ1 and shown here as a control (unaltered methylation in *FTSJ1* mutants).

**Table 2. FTSJ1 targets multiple human tRNAs at positions 32 and 34.**

| tRNA target | Humans | | | | Drosophila | | S. cerevisiae | | Mouse | |
|---|---|---|---|---|---|---|---|---|---|---|
| | Current RiboMethSeq | | Previous HPLC/MS | | Previous RiboMethSeq | | Previous HPLC and/or MS | | Previous HPLC/MS | |
| | N32 | N34 | N32 | N34 | N32 | N34 | N32 | N34 | N32 | N34 |
| Arg (UCG1)[a] | **Cm** | no | **Cm** (Li et al, 2020) | **no** | no | no | n.d. | n.d. | n.d. | n.d. |
| Arg (CCG) | **Um** | no | **Um, Cm** (Li et al, 2020) | **Cm**[#] (Li et al, 2020) | no | no | n.d. | n.d. | n.d. | n.d. |
| Arg (ACG) | **Cm*** | **I** | **Cm** (Li et al, 2020) | no | **Cm** (Angelova et al, 2020) | no | n.d. | n.d. | **Cm** (Nagayoshi et al, 2021) | n.d. |
| Arg (UCG2)[a] | **Cm** | no | n.d. | n.d. | n.d. | n.d. | n.d. | n.d. | n.d. | n.d. |
| Leu (CAG_CAA) 91%_9% | **Um** | **no** | no | **Cm**[@] (Kawarada et al, 2017; Li et al, 2020) | **Cm** (Angelova et al, 2020) | **Cm** (Angelova et al, 2020) | n.d. | | n.d. | **Um_** hm5Cm (Nagayoshi et al, 2021) |
| Leu (UAA) | no | **U?m*** | n.d. | no | no | no | **Cm** (Guy et al, 2012) | ncm5**Um** (Glasser et al, 1992; Guy et al, 2012) | **Cm** (Nagayoshi et al, 2021) | ncm5**Um** (Nagayoshi et al, 2021) |
| Leu (A/IAG) 76% | U/ψm | **I** | n.d. | no | no | no | n.d. | n.d. | **Ψm** (Nagayoshi et al, 2021) | n.d. |
| Leu (UAG) 24% | U/ψm | **no** | n.d. | no | no | no | n.d. | n.d. | **Um** (Nagayoshi et al, 2021) | n.d. |
| Phe (GAA) | **Cm*** | **Gm** | **Cm** (Guy et al, 2015; Li et al, 2020; Nagayoshi et al, 2021) | **Gm** (Guy et al, 2015; Li et al, 2020; Nagayoshi et al, 2021) | **Cm*** (Angelova et al, 2020) | **Gm** (Angelova et al, 2020) | **Cm** (Guy et al, 2012) | **Gm** (Guy et al, 2012) | **Cm** (Nagayoshi et al, 2021) | **Gm** (Nagayoshi et al, 2021) |
| Trp (CCA) | **Cm** | **Cm** | **Cm** (Guy et al, 2015; Li et al, 2020; Nagayoshi et al, 2021) | **Cm** (Guy et al, 2015; Li et al, 2020; Nagayoshi et al, 2021) | **Cm** (Angelova et al, 2020) | **Cm** (Angelova et al, 2020) | **Cm** (Guy et al, 2012) | **Cm** (Guy et al, 2012) | **Cm** (Nagayoshi et al, 2021) | **Cm** (Nagayoshi et al, 2021) |
| Gln (CUG_UUG) 92%_8% | **Cm** | no | **Cm** (Li et al, 2020) | n.d. | **Cm** (Angelova et al, 2020) | no | n.d. | n.d. | **Cm** (Nagayoshi et al, 2021) | |
| Gly (CCC) | **Um** | no | n.d. | no | no | no | n.d. | n.d. | n.d. | n.d. |
| Val (AAC_CAC_TAC) 73%_26%_1% | no | **I (AAC)** | **Cm** (Nagayoshi et al, 2021) | n.d. | **Cm** (Angelova et al, 2020) | no | n.d. | n.d. | **Cm** (Nagayoshi et al, 2021) | |
| Pro (AGG_CGG_UGG) 34%_23%_42% | **Um*** | **I (AGC)** | no | n.d. | n.d. | n.d. | n.d. | n.d. | n.d. | n.d. |

| tRNA target | Humans | | | | Drosophila | | S. cerevisiae | | Mouse | |
| | Current RiboMethSeq | | Previous HPLC/MS | | Previous RiboMethSeq | | Previous HPLC and/or MS | | Previous HPLC/MS | |
| | N32 | N34 | N32 | N34 | N32 | N34 | N32 | N34 | N32 | N34 |
|---|---|---|---|---|---|---|---|---|---|---|
| Cys (GCA_ACA) 97%_3% | Cm* | no | n.d. | n.d. | n.d. | n.d. | n.d. | n.d. | n.d. | n.d. |
| Met (CAU) non-FTSJ1 Target | no | Cm | no | Cm (Vitali & Kiss, 2019; Li et al, 2020) | no | no | no | no | n.d. | n.d. |

A summary of tRNA nucleotides revealed to date, including by the current study, as targets of human FTSJ1, and those targeted by Drosophila Trm7_32 and Trm7_34, and yeast Trm7 in the respective organisms. For the tRNA targets are given the isotype (determined by the bound amino acid) and the isoacceptor (determined by the ACL sequence). In blue are highlighted the studies done with the site-specific RiboMethSeq, and in grey, the ones done by mass spectrometry single-nucleotide analysis. n.d. stands for non-determined and indicates that the tRNA was not tested or if tested the data were not analysable. no stands for non-detected Nm. Cm, Gm, and Um stand for 2′-O-methylated, respectively, C, G, and U nucleotides. * indicates Nm RiboMethSeq detection by visual inspection of the raw read profile, not MethScore; see Fig S1D for an example. When several anticodon sequences are present for tRNA isoacceptors, proportion of every sequence in the healthy subject is indicated at the bottom. Cm# indicates Cm detection in Li et al (2020) that could be because of a high sequence similarity with another tRNAArg, tRNAArg(CCG)-2-1 containing a C32. The observed Cm decrease in FTSJ1 KO cells in this study may come from C32 of tRNAArg(CCG)-2-1 that was modified by FTSJ1 and not from the C34 level of tRNAArg(CCG). Cm@ indicates hm5Cm34 or f5Cm34 in tRNALeu(CAA) shown in Kawarada et al (2017). I stands for inosine (FTSJ1-independent). U?m* indicates clear FTSJ1 dependence; however, the exact nature of this modified U remains unknown. tRNAArg (UCG) and (CCG) have identical sequences but differ only at positions 32 and 34.
astands for UCG isodecoders (sequences in the Materials and Methods section). tRNALeu (A/IAG) and (UAG) are isoacceptors, they differ only by the N34 nucleotide, and both have Um32 (or ψm32).

tRNA regions impairs RT, thereby reducing the number of cDNAs spanning the ACL. Nevertheless, considering all these potential limitations when using only MethScore calculation, a visual inspection of raw cleavage profiles was performed (Fig S1D and Table 2) and revealed to be the most appropriate. When visualizing raw read count profile, reads' end number at position 33 (Cm$_{32}$) of tRNA$^{Phe(GAA)}$ was increased in FTSJ1-mutated cells (Fig S1D), indicating a loss of Cm$_{32}$ of tRNA$^{Phe(GAA)}$ in FTSJ1-mutated LCLs. Thus, using both MethScore (Fig 1) and visual inspection on all RiboMethSeq human tRNA sequences (Fig S1D) we were able to confirm known FTSJ1 tRNA targets and, importantly, discover new FTSJ1-dependent Cm$_{32}$/Um$_{32}$ modifications in tRNA$^{Gly}$, tRNA$^{Leu}$, tRNA$^{Pro}$, and tRNA$^{Cys}$ (see Table 2 for isoacceptor details). Unexpectedly, Um$_{34}$ in tRNA$^{Leu(UAA)}$ also demonstrated clear FTSJ1 dependence; however, the exact nature of this modified nucleotide remains unknown (Table 2). In contrast, the protection signal observed at position 32 in human tRNA$^{Ala(A/IGC)}$ is not FTSJ1-dependent and most likely results from ψm$_{32}$ (visible in HydraPsiSeq [Marchand et al, 2022] profiling) and not Um$_{32}$.

### FTSJ1 loss of function deregulates mRNA steady-state level

To obtain insights into the impact of FTSJ1 loss on gene expression, we performed a transcriptome analysis in patient and control LCLs. Transcript differential expression analysis shows that FTSJ1 dysfunction led to a deregulation of 686 genes (Table 3 and Fig S2A and B). This relatively low number is in agreement with a previous report showing 775 genes deregulated in human HeLa cells knock down for FTSJ1 (Trzaska et al, 2020), and with the 110 mRNAs deregulated in KD of one FTSJ1 Drosophila ortholog (Angelova et al, 2020).

Even though LCLs do not have a neural origin, analysis of the genes deregulated in affected individuals revealed a clear enrichment (FE = 7.9 with P-value = 7.44 × 10$^{-6}$ and FDR = 4.40 × 10$^{-3}$) in biological process gene ontology (GO) term corresponding to brain morphogenesis (Fig 2A). In addition, and similar to what we reported in a previous mRNA-seq analysis of Drosophila S2 cells knocked down for Trm7_34 (Angelova et al, 2020), five of the top 10 most enriched terms were related to mitochondrial biological processes. Also, in agreement with a recently described role of human FTSJ1 in translational control (Trzaska et al, 2020; Nagayoshi et al, 2021) and of yeast Trm7 in the general amino acid control pathway (Han et al, 2018), four biological processes related to translation were affected in FTSJ1-mutated LCLs (FE > 3.5, Fig 2A).

To strengthen the transcriptome analysis, we selected three representative and disease-relevant deregulated mRNAs based on their fold change level of expression and related involvement in brain or cancer diseases. Mutations in the human ZNF711 gene were previously reported to be involved in the development of ID (van der Werf et al, 2017). The mRNA-seq and qRT−PCR analyses showed a significant down-regulation of ZNF711 mRNA in FTSJ1 mutant LCLs when compared to control LCLs (Table 3 and Fig 2B). BTBD3 activity is known to direct the dendritic field orientation during development of the sensory neuron in the mouse cortex (Matsui et al, 2013) and to regulate mouse behaviours (Thompson et al, 2019). We found that BTBD3 mRNA was significantly up-regulated in both mRNA-seq and qRT−PCR analyses (Fig 2B). Lastly, SPARC (Tai & Tang, 2008) and more recently FTSJ1 (Holzer et al, 2019; He et al, 2020) gene product activities were proposed to be involved in both metastasis and tumour suppression. In the absence of FTSJ1, we could confirm that SPARC mRNA was significantly reduced (Table 3 and Fig 2B). Taken together, these results show deregulation of some mRNAs linked to cancer and brain functioning in FTSJ1 affected individuals' blood-derived LCLs.

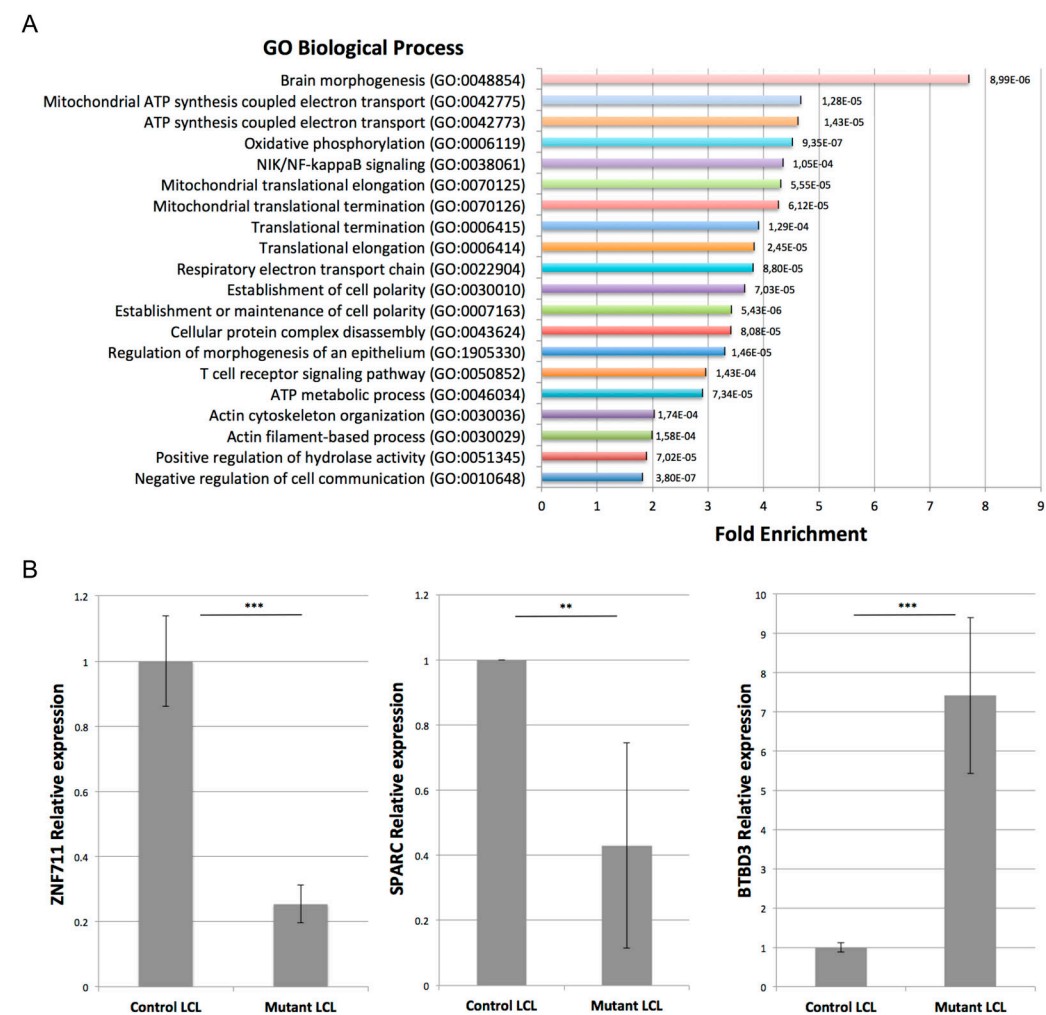

**Figure 2. *FTSJ1* loss of function leads to mRNA deregulation in XLID affected individuals' LCLs.**
**(A)** *FTSJ1* loss of function mRNA GO term. GO analysis of the 686 deregulated genes in *FTSJ1* function–deficient LCLs derived from XLID affected individuals (five mutants versus two control LCLs); *P*-values are indicated with error bars on the right of each box. The most enriched GO term is brain morphogenesis. GO analysis was performed using http://geneontology.org/. **(B)** qRT–PCR analysis confirms deregulation in *ZNF711*, *BTBD3*, and *SPARC* mRNA expression levels. Normalized to *GAPDH* steady-state levels. n > 3. *P*-values were calculated with a paired *t* test: **P < 0.01 and ***P < 0.001. WT values: mean of two control *FTSJ1* LCLs. Mutant values: mean of all (×5) *FTSJ1* mutant LCLs of this study, or two (LCL MM and LCL 65JW) for *ZNF711* qRT-PCR.

## *FTSJ1* loss of function affects the miRNA population

Our previous work on the *Drosophila* homologs of *FTSJ1*, *Trm7_32* and *Trm7_34*, showed that their loss of functions led to perturbations in the small non-coding RNA gene silencing pathways, including the miRNA population (Angelova et al, 2020). To address whether such small RNA perturbations are conserved in XLID affected individuals, we performed small RNA sequencing on the five LCLs carrying *FTSJ1* loss of function variants compared with the four LCLs from control individuals. The principal component analysis of the different *FTSJ1* loss of function cell lines shows a high similarity and thus clusters on the principal component analysis plot, whereas the WT lines were more dispersed, possibly explained by their geographic origins (Fig S3A). The DESeq2 differential expression analysis showed statistically significant deregulation of 36 miRNAs when comparing *FTSJ1* mutants to control LCLs. 17 miRNAs were up- and 19 down-regulated (Figs 3A and S3B and log₂ FC and

adjusted *P*-values in Table S1). Importantly, as already reported in *Drosophila* (Angelova et al, 2020), the global miRNA distribution was not drastically affected, thus ruling out general involvement of FTSJ1 in miRNA biogenesis.

Next, we sought for possible links between the 36 significantly deregulated miRNAs in *FTSJ1* mutant cells and neuronal functions or neurodevelopmental disorders. Interestingly, 21 of these miRNAs were already identified in other small RNA-seq studies performed in the context of brain diseases such as epilepsy, and Parkinson's and Alzheimer's diseases (Lau et al, 2013; Kretschmann et al, 2015; Ding et al, 2016; Roser et al, 2018). In addition, 29 of the deregulated miRNAs were linked to different types of cancers (Lund, 2010; Watahiki et al, 2011; Li et al, 2015; Khuu et al, 2016; Yang et al, 2017; Jiang et al, 2018), including 21 involved specifically in brain-related cancers, mostly in glioblastoma (Gillies & Lorimer, 2007; Shi et al, 2008; Lund, 2010; Conti et al, 2016) (Fig 3B and Table 4).

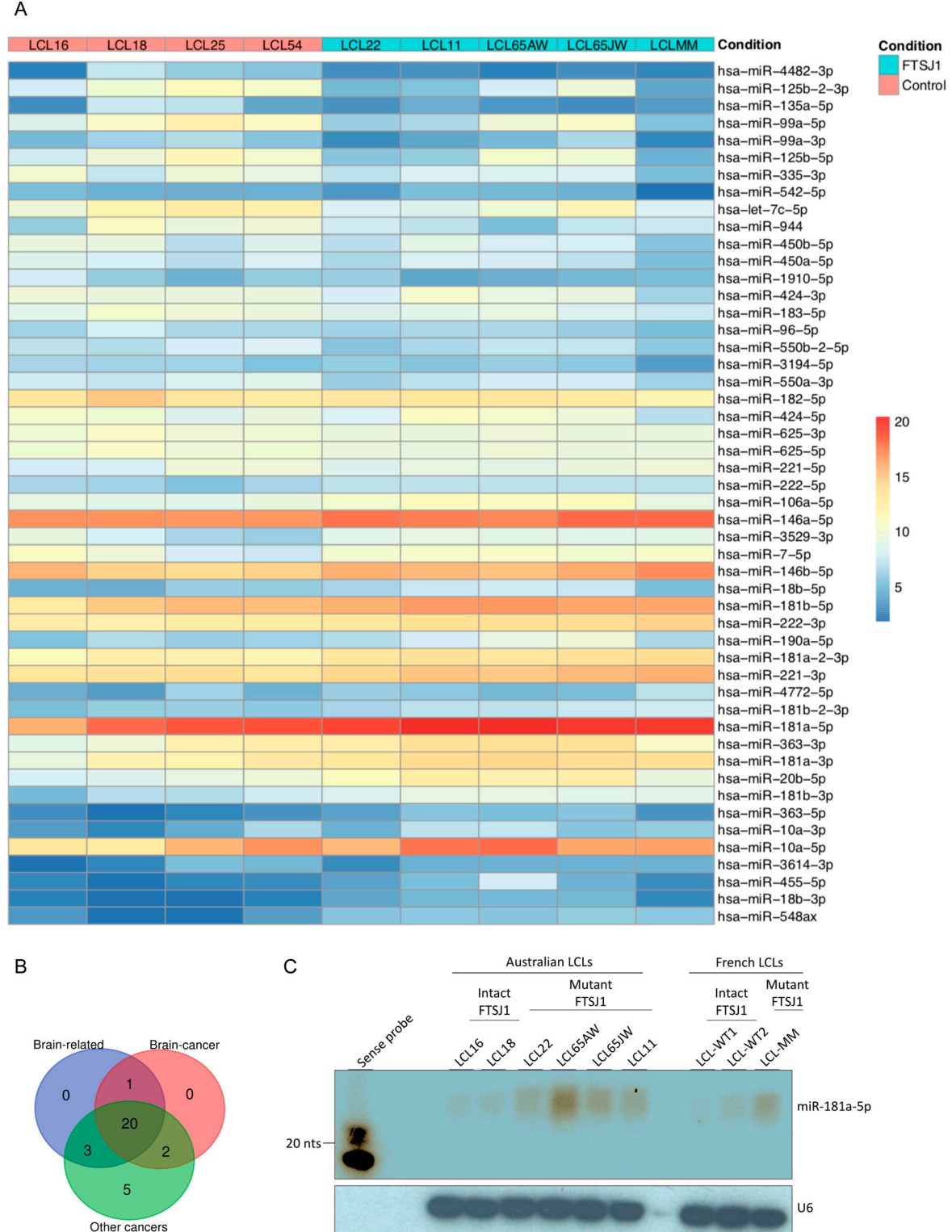

**Figure 3. *FTSJ1* loss of function leads to miRNA deregulation in XLID affected individuals' LCLs.**
**(A)** Heat map generated using the pheatmap package in R showing the 50 best deregulated miRNAs in *P*-values, and sorted fold change from most down-regulated (blue) to most up-regulated (red) is represented in two experimental conditions: *FTSJ1* loss of function LCLs (blue turquoise) compared with controls LCLs (pink). Condition points to the FTSJ1 LCL status, WT (control) or mutated for the *FTSJ1* gene (FTSJ1). The data come from normalized and variance-stabilizing transformed read counts using the DESeq2 package in R. **(B)** Bibliographic search (Table 4) of the miRNAs deregulated in *FTSJ1* loss of function LCLs reveals evidence for many of them as being implicated in cancers or brain development and brain diseases. The number of miRNAs related to brain, cancer, and brain–cancer specifically is indicated

To strengthen the small RNA-seq data, four hemizygous *FTSJ1* LCLs (control) and five LCL mutants for *FTSJ1* were analysed by Northern blotting with a specific probe complementary to *miRNA-181a-5p*. We selected this miRNA as it was highly up-regulated in our small RNA-seq analysis and it was previously reported to be involved in vascular inflammation and atherosclerosis (Su et al, 2019), and expressed in neuronal cells in mammals (Dostie et al, 2003). One clear hybridization signal was observed in all *FTSJ1* mutant LCLs corresponding to mature *miRNA-181a-5p* (Fig 3C). In contrast, the four control LCLs show no or weak signal even after image over-exposure (Fig 3C). Together, these results demonstrate that *FTSJ1* loss of function affects specifically the steady-state levels of some miRNA and suggest that the deregulation of miRNA-mediated gene silencing observed in *FTSJ1* mutant LCLs was not caused by a global failure in miRNA biogenesis (Figs 3A and S3B and Table S1).

### *FTSJ1* mutation perturbates the silencing activity of *miR-181a-5p* miRNA

As some of the *FTSJ1*-deregulated miRNAs and mRNAs were implicated in similar biological processes such as cancer and brain function, we wondered whether there were some miRNA::mRNA pairs that could be involved in these commonly deregulated processes. Using miRNet 2.0 (Chang et al, 2020), we performed a bioinformatics cross-analysis of the small RNA-seq and mRNA-seq datasets. We found a subset of *FTSJ1*-deregulated miRNAs that were previously shown to modulate some of the *FTSJ1*-deregulated mRNAs. For instance, the *SPARC* mRNA is an experimentally confirmed target of *miR-10a-5p* (Bryant et al, 2012; Wang et al, 2020). This result thus suggests that *SPARC* mRNA down-regulation observed in *FTSJ1* mutants may be due to its increased silencing by the up-regulated *miR-10a-5p*. This cross-analysis also revealed that the *BTBD3* gene is potentially targeted by *miR-181a*-5p (He et al, 2015), the two of which were up-regulated in XLID affected individuals' blood-derived LCLs (Fig 3A and C and Table 4), implicating a possible connection between them that differs from the canonical miRNA silencing pathway. LCLs are known to be hardly transfectable (Nagayoshi et al, 2021); however, *miR-181a-5p* and *BTBD3* are expressed similarly in HeLa cells (Fig S4A). Thus, by mimicking *miR-181a-5p* expression or repression, we show that *miR-181a-5p* silences *BTBD3* in HeLa cells (Fig S4B), strongly suggesting that *BTBD3* mRNA is a bona fide target of *miR-181a-5p*. Strikingly, in *FTSJ1* mutant cells, the silencing activity of *miR-181a-5p* on *BTBD3* is compromised in both HeLa and LCL. Interestingly, despite the fact that 39 *ZNF* mRNAs were found potentially regulated by *miR-181a-5p* (Table 4 and [He et al, 2015]) and the over-representation of this miRNA in *FTSJ1* mutant (Fig 3A and C and Table S1), no evidence of miRNA regulation was yet found for *ZNF711*, a gene previously reported to be involved in the development of ID (van der Werf et al, 2017).

### FTSJ1 is involved in human neuronal morphology during development

The loss of *FTSJ1* in humans gives rise to XLID, yet the underlying mechanism is still unclear. Defects in both neuronal morphology (Chen et al, 2021) and behaviour (Jensen et al, 2019) have been reported in patients affected by a wide range of ID disorders, with a variety of genetic aetiologies and their corresponding mouse models. To address whether the loss of human *FTSJ1* also affects neuronal morphology, we altered FTSJ1 activity using 2,6-diaminopurine (DAP) (Trzaska et al, 2020; Palma & Lejeune, 2021) in human NPCs. DAP is a recently discovered drug that binds to FTSJ1 and inhibits its methyltransferase activity (Trzaska et al, 2020; Palma & Lejeune, 2021). Immunostainings were performed for Sox2, a transcription factor expressed in NPCs, and doublecortin (DCX), a microtubule-associated protein expressed in differentiating NPCs or immature neurons, reflecting neurogenesis. Importantly, the DAP treatment did not significantly affect the differentiation of the NPCs (DCX–) to immature neurons (DCX+) (Fig 4A). This is in agreement with previous reports showing the absence of severe brain morphological defects in mice mutated for *Ftsj1* (Jensen et al, 2019; Nagayoshi et al, 2021). However, DCX-positive cells treated with 100 $\mu$M DAP showed a 25% increase in the number of interstitial protrusions, likely filopodia, on their neurites compared with the smoother appearance of the neurites of untreated control cells (Fig 4B and C). These spines' morphological defects on DAP-treated DCX+ cells are reminiscent of those observed on mature neurons from mutant mice of the fragile X mental retardation protein (*Fmr1*) (Braun & Segal, 2000) and from human patients' brains that suffer from the fragile X syndrome (Irwin et al, 2000). Furthermore, similar findings were recently reported in mouse brains mutated for *Ftsj1* (Nagayoshi et al, 2021), suggesting that this is a conserved phenotypic consequence of the loss of *FTSJ1*.

### *Drosophila FTSJ1* ortholog is involved in neuronal morphology during development

To further address whether the control of neuron morphology by FTSJ1 is a conserved feature across evolution, we dissected the neuromuscular junctions (NMJs) of *Drosophila* larvae carrying mutations in the orthologs of the *FTSJ1* gene, and larvae fed with DAP (Trzaska et al, 2020; Palma & Lejeune, 2021). Examination of the NMJs in *Trm7_32* and *Trm7_34* double homozygous mutant larvae or larvae fed with DAP revealed a significant synaptic overgrowth when compared to control larvae (Fig 5). Furthermore, as observed for the human NPC treated with DAP (Fig 4B and C), the neurite branching was strongly increased in both double mutant and DAP-fed larvae (Fig 5). However, the overall length of the axons was not significantly altered. These results indicate that *Drosophila* FTSJ1s, like human FTSJ1, control neuronal morphology.

respectively in the blue, green, and red circles. The Venn diagram was generated by http://bioinformatics.psb.ugent.be/webtools/Venn/. **(C)** Northern blot analysis with ³²P-labelled probe specific for *hsa-miR-181a-5p* confirms the up-regulation of this miRNA in *FTSJ1* loss of function condition already detected by small RNA-seq analysis. A ³²P-labelled probe specific for human U6 RNA was used to assess equal loading on the blot.

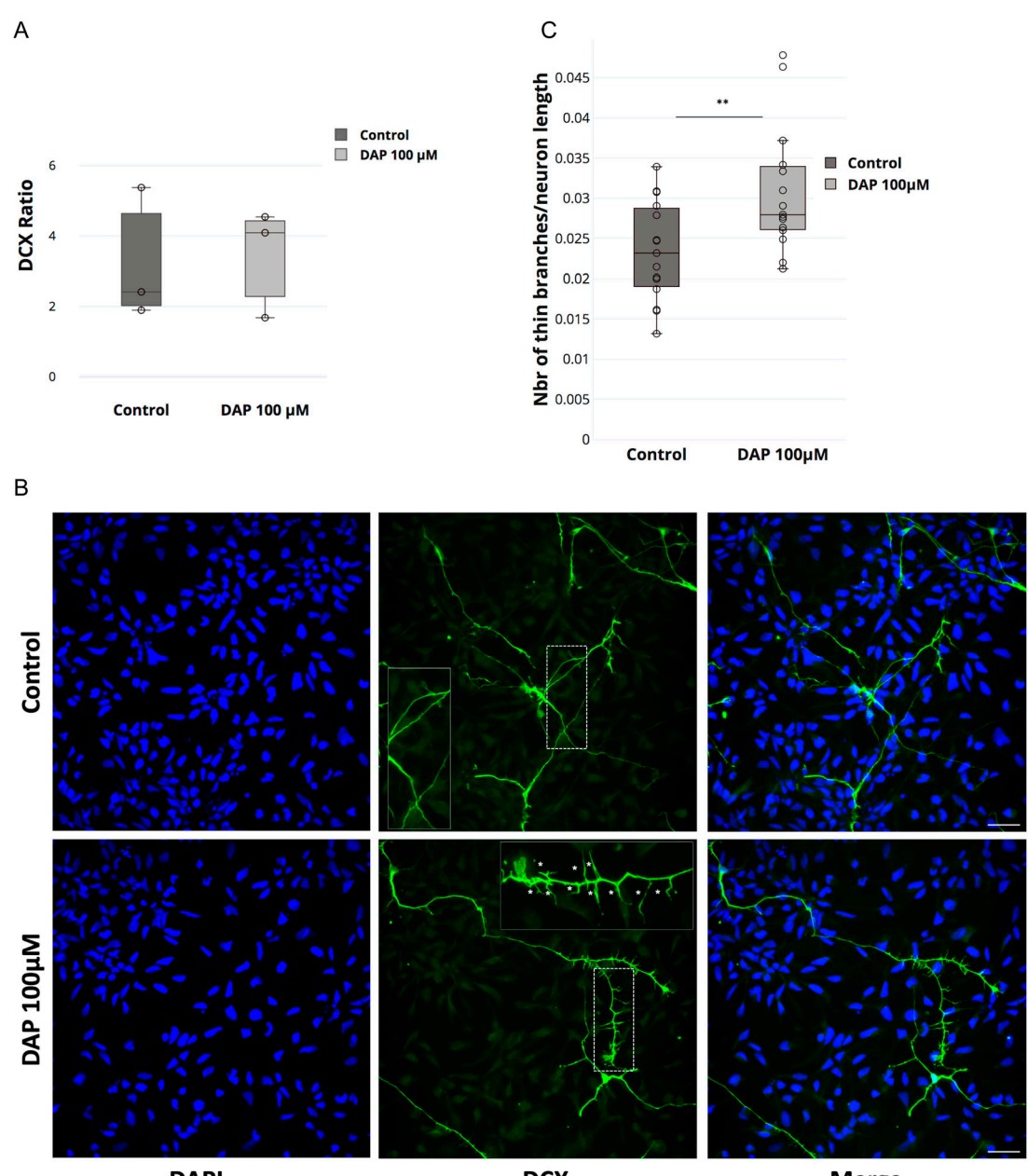

**Figure 4. FTSJ1 depletion affects human neurons' spine morphology.**
**(A)** DAP-induced FTSJ1 inhibition does not affect human NPC to immature neuron differentiation. Immunostainings for DCX and SOX2 were performed on human iPSC-derived NPCs treated with either 100 $\mu$M DAP or equal volume of $H_2O$ for 24 h. Cells were numbered on microscopy acquisitions, and the ratio of DCX-expressing cells over total cell number was calculated and expressed in fold change. Error bars represent SD of three independent experiments; n.s., not significant (over 1,400 cells numbered for a single experiment). **(B)** Lower panel: human NPCs inhibited for FTSJ1 with 100 $\mu$M DAP for 24 h (DAP 100 $\mu$M) present an increased number of neurite spines during NPC to immature neuron differentiation. DCX protein expressed in immature neurons is marked in green (DCX). Dashed white line represents the *zoom-in* zone depicted in the top right corner with a continuous white line. White stars (*) in the magnified inset point to the fine spine neurites. Upper panel: untreated NPCs (control). Nuclear staining was performed using DAPI depicted in blue (DAPI). **(C)** Quantification of thin spines of DCX-positive cells ((B) above). Thin projections were numbered and normalized over the total length of the immature neurons as traced and measured by Simple Neurite Tracer (Fiji plugin). Quantifications were carried out on five acquisitions for each experiment (control and DAP 100 $\mu$M) (>40 branches/acquisition on average). White scale bar: 30 $\mu$m. Aggregate of three independent experiments. Wilcoxon–Mann–Whitney's test, **$P$ = 0.0098.

## Reward learning requires FTSJ1 activity in *Drosophila*

*FTSJ1* loss of function affected individuals suffer from significant limitations both in intellectual functioning and in adaptive behaviour. Similar phenotypes including impaired learning and memory capacity were recently observed in *Ftsj1* KO mice that also present a reduced body weight and bone mass, and altered energy metabolism (Jensen et al, 2019; Nagayoshi et al, 2021). In flies, we

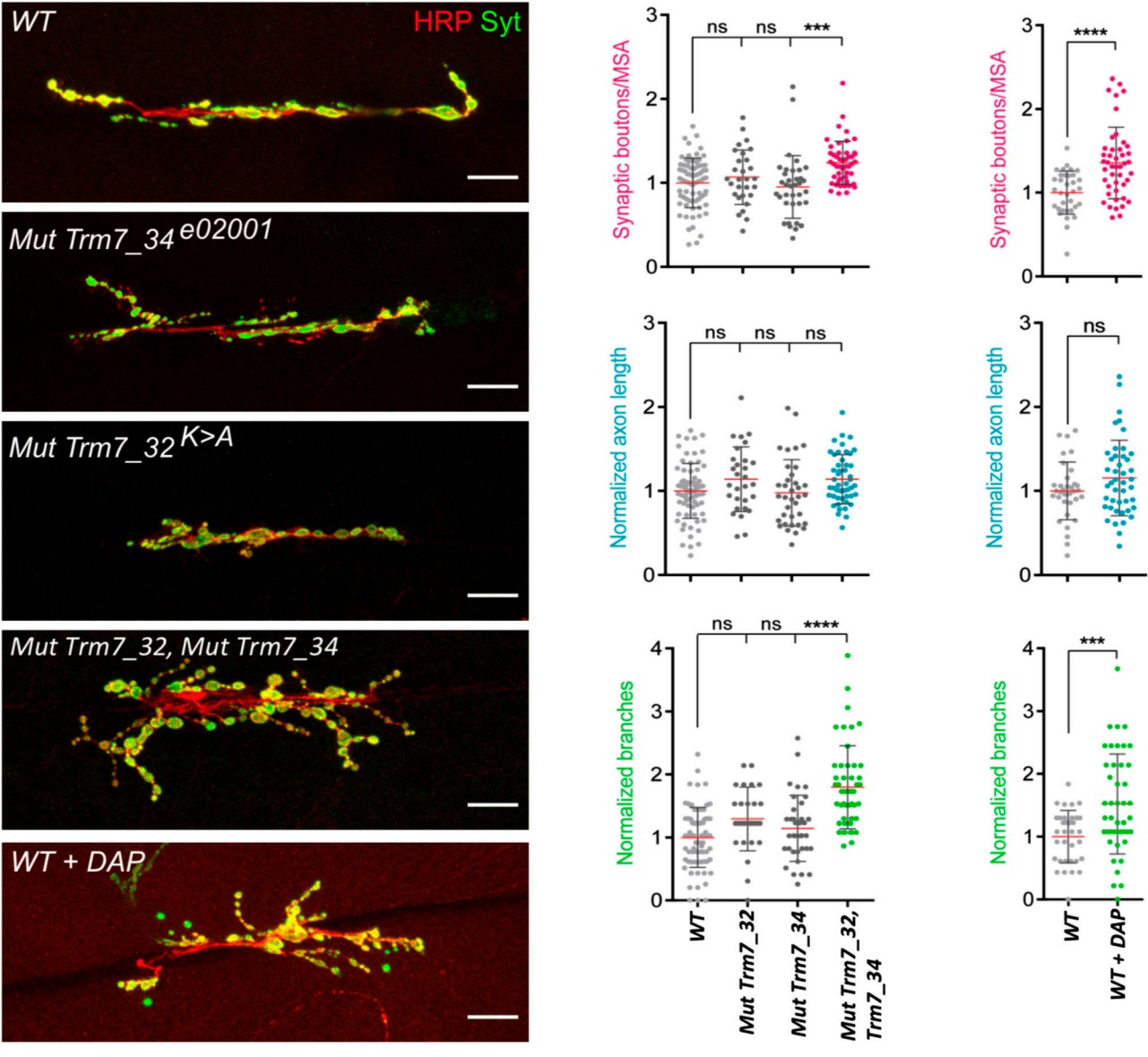

**Figure 5.  FTSJ1-dependent Nm regulates axonal morphology in the *Drosophila* nervous system.**
Left panel: representative confocal images of muscle-6/7 NMJ synapses of larval abdominal hemisegments A2–A3 for the indicated genotypes labelled with anti-synaptotagmin (green) and HRP (red) to reveal the synaptic vesicles and the neuronal membrane. White scale bar: 20 $\mu m$. Right panel: quantification of normalized bouton number (total number of boutons/muscle surface area [$\mu m^2$ × 1,000]) (top), normalized axon length (middle), and normalized branching (bottom) of NMJ 6/7 in A2–A3 of the indicated genotypes. Bars show the mean ± SEM. Multiple comparisons were performed using one-way ANOVA with a post hoc Sidak–Bonferroni correction (n.s., not significant; *$P < 0.05$; ***$P < 0.001$; and ****$P < 0.0001$). Numbers of replicated neurons (n) are as follows: 74 for WT; 36 for *Trm7_32*; 29 for *Trm7_34*; 48 for *Trm7_32*; *Trm7_34*; and 34 for WT untreated and 45 for WT treated with DAP. *Canton-S* larvae were used as WT control.

recently showed that the loss of *FTSJ1* orthologs causes reduced lifespan and body weight, and locomotion defects (Angelova et al, 2020).

To address whether fly memory was also altered in these mutants, we applied an appetitive conditioning assay. We found that short-term memory of single *Trm7_34* or *Trm7_32* and double *Trm7_34;Trm7_32* heterozygous mutant flies was indistinguishable from that of wild-type controls (Fig 6A). However, long-term memory (LTM) was significantly impaired in all of these three

mutant combinations (Fig 6B). Importantly, naive heterozygous mutant flies detected sugar properly and behave normally when exposed to repellent odours used in the olfactory memory assay (Fig 6C and D), suggesting that the LTM defect was not due to a confounding alteration of sensory abilities. Thus, these results indicate that the *Drosophila FTSJ1* orthologs *Trm7_34* and *Trm7_32* have a specific function in LTM, and importantly demonstrate clearly that both tRNA Nm32 and Nm34 modifications have function in long-term memory.

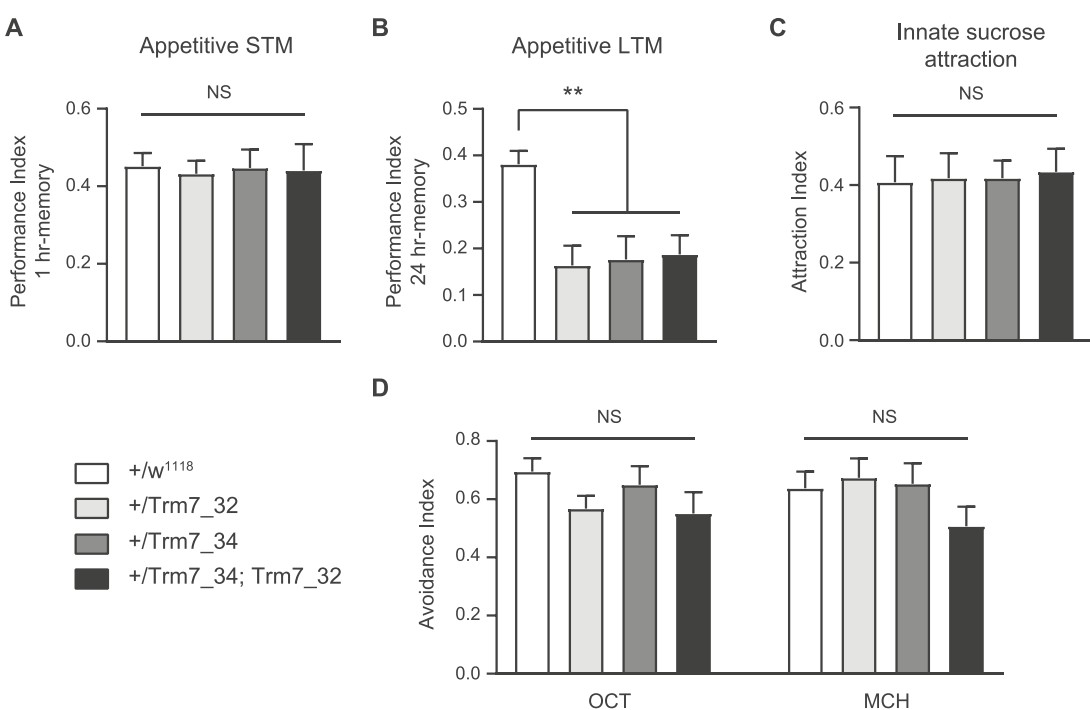

**Figure 6.** *Drosophila FTSJ1 ortholog Trm7_34 mutants are defective for appetitive long-term memory.*
Behavioural performances are reported as the mean±SEM. Statistical significance was tested with a one-way ANOVA followed by Tukey's post hoc pairwise comparisons. Asterisks on the barplots indicate the level of significance of the pairwise comparison with control. The *P*-value indicated in the legend corresponds to the output of the ANOVA. **(A)** Flies were starved on mineral water for 21 h and then trained with an appetitive associative olfactory learning protocol (odour paired with sucrose ingestion). Short-term memory performance was measured 1 h after learning. The short-term memory score of single *Trm7_32* (+/Trm7_32) and *Trm7_34* (+/Trm7_34), and double *Trm7_32*; *Trm7_34* (Trm7_32; Trm7_34) heterozygous mutant flies was not different from their genotypic controls (+/w[1118]) (n = 12 per condition; *P* = 0.99). **(B)** Flies were starved on mineral water for 21 h and then trained with an appetitive associative olfactory learning protocol (odour paired with sucrose ingestion). Long-term memory performance was measured 24 h after learning. The long-term memory score of single *Trm7_32* (+/Trm7_32) and *Trm7_34* (+/Trm7_34), and double *Trm7_32*; *Trm7_34* (Trm7_32; Trm7_34) heterozygous mutant flies was severely impaired as compared to their genotypic controls (+/w[1118]) (n = 16–19 per condition; *P* = 0.0007). **(C)** Flies were starved on mineral water for 21 h, and their attraction to sucrose was then measured. The innate sucrose preference of single *Trm7_32* (+/Trm7_32) and *Trm7_34* (+/Trm7_34), and double *Trm7_32*; *Trm7_34* (Trm7_32; Trm7_34) heterozygous mutant flies was not different from their genotypic controls (+/w[1118]) (n = 14 per condition; *P* = 0.99). **(D)** Flies were starved on mineral water for 21 h, and their avoidance to the odorants used in the olfactory memory assays, 3-octanol (OCT) and 4-methylcyclohexanol (MCH), was then measured. The innate odour avoidance of single *Trm7_32* (+/Trm7_32) and *Trm7_34* (+/Trm7_34), and double *Trm7_32*; *Trm7_34* (Trm7_32; Trm7_34) heterozygous mutant flies was not different from their genotypic controls (+/w[1118]) (n = 10 per condition; OCT: *P* = 0.26; MCH: *P* = 0.28).

# Discussion

In this study, we characterized at the molecular and cellular levels the effect of *FTSJ1* loss of function in human cells. We used the innovative RiboMethSeq method to analyse the Nm status from five patients carrying a distinct loss of *FTSJ1* functions, which led us to the identification of new human FTSJ1 tRNA targets. Furthermore, we identify specific transcripts and miRNA that are misregulated in the absence of FTSJ1, which may contribute to the FTSJ1 pathologies, and suggest potential cross-regulation among them. Lastly, we show for the first time that the lack of FTSJ1 alters the morphology of human neurons, a phenotype that is conserved in *Drosophila* and is associated with long-term memory deficits.

The power of the RiboMethSeq approach is that it allows to analyse the Nm status of the totality of transcribed tRNA species and not only selected tRNAs based on the prior but incomplete knowledge of FTSJ1 targets. Furthermore, this approach covers the whole tRNA-ome and thus can identify variations in Nm at the single nucleotide resolution, which is very useful to distinguish

tRNA isoacceptors for instance that differ by only a few nucleotides. Our results from the RiboMethSeq performed on patient and control LCLs confirmed the already known human tRNA targets of FTSJ1. For instance, $Cm_{32}$ and $Cm_{34}$ of tRNA[Trp(CCA)], and position 34 in tRNA[Phe(GAA)] and tRNA[Leu(CAG)] were validated by our approach. Only $Cm_{32}$ of tRNA[Phe(GAA)], which is a well-known target of FTSJ1, could not be validated at the first glance. The analysis of this position is challenging because of low read numbers necessary for its quantification. This is the result of two confounding factors. On the one hand, the calculation of MethScores (Fig 1A) is based on the two neighbouring nucleotides (Marchand et al, 2016). Because FTSJ1 deposits Nm at both 32 and 34 positions in tRNA[Phe], the calculated MethScore at position 32 is affected when position 34 of the same tRNA is also Nm-modified. On the other hand, we previously reported that tRNA[Phe(GAA)] ACL positions are challenging to detect because of the specific hyper-modification on position 37 of tRNA[Phe] (Angelova et al, 2020). Indeed, $o2yW_{37}/m^1G_{37}$ impairs RT, thereby reducing the number of cDNAs spanning the ACL. Nevertheless, deeper visual inspection of the raw read profile shows that

Nm at position 32 was indeed lost in *FTSJ1*-mutated cells when compared to control LCL (Fig S1D), confirming the previous reports.

Importantly, we confirmed recent (tRNA$^{Arg(UCG)}$ and tRNA$^{Gln(CUG)}$) and identified novel (tRNA$^{Gly(CCC)}$, tRNA$^{Leu(UAA)}$, tRNA$^{Pro}$, and tRNA$^{Cys(GCA)}$; Table 2) tRNA targets for human FTSJ1. In the case of tRNA$^{Arg(UCG)}$, we confirmed not only a new target for FTSJ1, but also a modification, which was not previously reported in MODOMICS but only recently in the HEK293 *FTSJ1* CRISPR mutant (Li et al, 2020). Indeed, C$_{32}$ is known to be m$^3$C and not Nm-modified for the two other isoacceptors (tRNA$^{Arg(CCU)}$ and tRNA$^{Arg(UCU)}$) (Boccaletto et al, 2018). Similarly, there was no evidence for a human Cm$_{32}$ tRNA$^{Gln(CUG)}$ and only the other isoacceptor tRNA$^{Gln(UUG)}$ was reported in MODOMICS as 2′-O-methylated at C$_{32}$. Still, Cm$_{32}$ on tRNA$^{Gln(CUG)}$ was recently discovered as a target of *Drosophila* Trm7_32 (Angelova et al, 2020). Among the newly uncovered FTSJ1 targets in this study, Um$_{32}$ tRNA$^{Gly(CCC)}$ was the only one that has been reported in MODOMICS; however, the enzyme responsible for this modification was yet unknown. Our results demonstrate that FTSJ1 is the dedicated human Nm-MTase that installs Um$_{32}$/Cm$_{32}$ and Cm$_{34}$/Um$_{34}$/Gm$_{34}$ residues on human tRNAs.

Our transcriptomic analysis also highlighted novel transcripts and miRNA targets that may play important roles in the development of the diseases. For instance, we found 36 differentially expressed miRNAs, most of which were already associated with brain diseases and functioning and/or cancer development. Strikingly, the most prevalent associated cancer types were the ones related to the brain tissues. Consistently with the post-transcriptional regulatory role of miRNAs, we also found through mRNA-seq an enrichment of brain morphogenesis–related mRNAs differentially expressed in *FTSJ1* loss of function when compared to control LCLs. Interestingly, a cross-analysis of these two RNA-sequencing experiments revealed potential miRNA::target mRNA couples among the deregulated RNA populations. This is indicative of possible miRNA silencing changes in the absence of FTSJ1, similar to what we report earlier in *Drosophila FTSJ1* mutant orthologs. The predicted miRNA::mRNA couples need to be further validated individually in neuronal tissues, although their report from *miRnet* database (Chang et al, 2020) already includes experimental evidence on the miRNA::mRNA regulation, particularly for *BTBD3* and *SPARC* mRNAs (Bryant et al, 2012; He et al, 2015; Wang et al, 2020). In addition to the reported prediction (He et al, 2015), we show that *BTBD3* is a bona fide *miR-181a-5p* target. Surprisingly, both *BTBD3* and *miR-181a-5p* were up-regulated in *FTSJ1*-depleted patient cells. Although the precise mechanism is not known yet, our results suggest that Nm-MTase genes could act upstream of small RNA biogenesis and function through transcriptional down-regulation of Argonaute mRNA in *Drosophila FTSJ1* mutants (Angelova et al, 2020) and in human cells (not shown). On the contrary, tRNA fragment (tRF) abundance seen in *FTSJ1* mutant fly (Angelova et al, 2020) and mice (Nagayoshi et al, 2021) can associate with Dicer, Argonaute, and Piwi proteins, thus affecting their silencing function. Such tRF-mediated titration of proteins away from canonical substrates has been previously reported in *Drosophila* and human cell lines (Durdevic et al, 2013; Goodarzi et al, 2015).

Affected individuals carrying mutations in *FTSJ1* suffer from XLID (Freude et al, 2004; Ramser et al, 2004; Guy et al, 2015), but the mechanism underlying this pathology has remained elusive. A recent report from Nagayoshi et al added some insight by showing that *Ftsj1* loss of function in mice provokes dendritic spine overgrowth at hippocampus and cortex neurons (Nagayoshi et al, 2021), suggesting that a similar alteration of neuron morphology may exist in human patients, which might impair their functioning. Indeed, we observed long, thin protrusion in human neurons affected for FTSJ1 activity. These protrusions are very similar in size and shape to the dendritic spines observed in hippocampus and cortex neurons of *Ftsj1* loss of function mice (Nagayoshi et al, 2021). A similar observation was also described earlier for *Fmr1* mutant mice (Braun & Segal, 2000) and *FMRP* human affected individuals' brains suffering from ID (Irwin et al, 2000). More examples of improper neuron morphology and in particular spine immaturity were found in additional gene loss of functions causative of ID (Levenga & Willemsen, 2012). This suggests that the lack of proper neuronal morphology may be a common feature of ID. More work will be required to address how these changes in spine arborization occur in the absence of FTSJ1 and how this translates into the disease. Interestingly in this study, we found that *BTBD3* mRNA is significantly up-regulated in *FTSJ1*-mutated LCLs. Because BTBD3 controls dendrite orientation in mammalian cortical neurons (Matsui et al, 2013), it will be an interesting target to further characterize in the context of FTSJ1 ID pathology.

A synaptic overgrowth was also observed in *Drosophila*, indicating that this function of FTSJ1 is conserved across evolution. In addition, we found that the long-term memory but not the short-term was significantly altered in the absence of FTSJ1 in flies. This is consistent with the learning deficits observed in mice and humans. In contrast to human FTSJ1 and the yeast ortholog TRM7, *Drosophila* uses two distinct paralogs to methylate positions 32 and 34, respectively, on the tRNA ACL. Interestingly, we found that the lack of both Trm7_34 and Trm7_32 had an effect on long-term memory, suggesting that the methylations at wobble positions 34 and 32 are critical for this function. However, the lack of both modifications (as in mammalian *Ftsj1* mutant) is not cumulative regarding the memory deficit (Fig 6). This last observation is strongly supported by the affected human individual who harbours a missense variant (p.Ala26Pro, LCL22 in this study), resulting in loss of Gm$_{34}$, but not of Cm$_{32}$, in human tRNA$^{Phe}$ (Guy et al, 2015). Further studies should aim to understand how the loss of methylation at these ACL positions affects the learning and memory functions.

The heterogeneity of ID makes it extremely challenging for genetic and clinical diagnoses (Ilyas et al, 2020). Our RiboMethSeq and transcriptomic approaches performed on XLID affected individuals have with high confidence extended the panel of FTSJ1's targets. Because our investigation was carried out on LCLs derived from the blood of affected individuals, our resource provides potential new biomarkers for diagnosis of FTSJ1-related ID in future. For instance, *miR-181a-5p*, which is detected only in patient-derived blood cells, constitutes already a good candidate for such purpose. Therefore, our study highlights the usefulness of companion diagnostics in clinical settings, in addition to exome sequencing, for potential discovery of prognostic markers of complex diseases.

**Table 3.** *FTSJ1* loss of function leads to mRNA deregulation in XLID affected individuals' LCLs.

| # | Symbol | baseMean_mutant | baseMean_wt | log₂FoldChange_Mutant_versus_WT | padj | # | Symbol | baseMean_mutant | baseMean_wt | log₂FoldChange_Mutant_versus_WT | padj |
|---|--------|-----------------|-------------|--------------------------------|------|---|--------|-----------------|-------------|--------------------------------|------|
| 1 | SASH1 | 1002.73 | 7.65 | 7.33 | $2.69 \times 10^{-41}$ | 36 | RNASE6 | 639.96 | 1645.30 | −1.72 | $1.43 \times 10^{-7}$ |
| 2 | FCRL4 | 515.44 | 7.09 | 6.05 | $2.08 \times 10^{-26}$ | 37 | CD38 | 3629.12 | 297.01 | 2.52 | $1.60 \times 10^{-7}$ |
| 3 | GSTT1 | 381.66 | 1.49 | 8.82 | $1.81 \times 10^{-18}$ | 38 | LOC728640 | 2914.46 | 2322.11 | 0.80 | $2.04 \times 10^{-7}$ |
| 4 | PPP1R21 | 5078.43 | 6380.13 | −1.06 | $1.92 \times 10^{-17}$ | 39 | APBB2 | 1332.24 | 3371.16 | −2.21 | $2.18 \times 10^{-7}$ |
| 5 | TINAG | 522.56 | 2.62 | 7.59 | $5.23 \times 10^{-17}$ | 40 | USMG5 | 7566.88 | 6750.46 | 0.76 | $2.45 \times 10^{-7}$ |
| 6 | ADCY6 | 544.04 | 17.61 | 3.38 | $1.20 \times 10^{-16}$ | 41 | FBN2 | 638.23 | 76.83 | 3.67 | $2.64 \times 10^{-7}$ |
| 7 | DSC2 | 990.21 | 134.24 | 3.01 | $9.55 \times 10^{-15}$ | 42 | HTR7 | 2.11 | 273.06 | −21.67 | $3.12 \times 10^{-7}$ |
| 8 | IL17RB | 3188.28 | 616.47 | 3.45 | $1.22 \times 10^{-14}$ | 43 | ALOX5 | 1890.29 | 5091.55 | −2.91 | $3.96 \times 10^{-7}$ |
| 9 | ABCA12 | 1267.89 | 3695.42 | −3.75 | $1.49 \times 10^{-14}$ | 44 | DDX60L | 962.23 | 110.32 | 2.42 | $5.75 \times 10^{-7}$ |
| 10 | JAZF1 | 333.25 | 17.53 | 5.14 | $1.99 \times 10^{-14}$ | 45 | B3GALNT1 | 617.28 | 26.67 | 4.51 | $8.73 \times 10^{-7}$ |
| 11 | TNRC6C | 1612.67 | 274.90 | 2.90 | $3.30 \times 10^{-14}$ | 46 | COX7B | 12243.87 | 9892.47 | 0.64 | $1.05 \times 10^{-6}$ |
| 12 | SYNE1 | 4579.60 | 5860.78 | −0.97 | $2.35 \times 10^{-13}$ | 47 | CBLB | 3203.88 | 5817.18 | −1.72 | $1.33 \times 10^{-6}$ |
| 13 | CPXM1 | 2071.71 | 6.69 | 8.34 | $2.34 \times 10^{-12}$ | 48 | PAPLN | 1212.67 | 3785.94 | −1.77 | $1.35 \times 10^{-6}$ |
| 14 | FNIP2 | 787.53 | 60.32 | 2.80 | $8.93 \times 10^{-12}$ | 49 | ANKRD26P3 | 561.85 | 2.58 | 8.71 | $1.79 \times 10^{-6}$ |
| 15 | CDH2 | 1169.70 | 45.93 | 5.50 | $1.06 \times 10^{-10}$ | 50 | ACVR2B | 355.06 | 771.16 | −1.78 | $1.88 \times 10^{-6}$ |
| 16 | TBX15 | 2674.86 | 22.11 | 5.58 | $1.52 \times 10^{-10}$ | 51 | RBPMS | 341.58 | 0.52 | 8.65 | $2.01 \times 10^{-6}$ |
| 17 | C14orf105 | 2783.34 | 168.09 | 3.35 | $4.51 \times 10^{-10}$ | 52 | PSMD7 | 34420.17 | 28659.74 | 0.64 | $2.07 \times 10^{-6}$ |
| 18 | AMPD3 | 1793.58 | 4186.29 | −2.29 | $5.50 \times 10^{-10}$ | 53 | MPHOSPH8 | 33819.66 | 28333.28 | 0.63 | $2.38 \times 10^{-6}$ |
| 19 | GAS2 | 2013.39 | 25.82 | 5.45 | $7.09 \times 10^{-10}$ | 54 | CTSW | 110.71 | 8.47 | 5.36 | $2.71 \times 10^{-6}$ |
| 20 | EVC | 293.08 | 3187.02 | −6.61 | $7.09 \times 10^{-10}$ | 55 | MYO9B | 10534.78 | 14242.52 | −0.53 | $2.84 \times 10^{-6}$ |
| 21 | TNFAIP2 | 1331.48 | 3608.85 | −1.82 | $1.15 \times 10^{-9}$ | 56 | IQGAP2 | 6537.41 | 9197.51 | −1.20 | $3.22 \times 10^{-6}$ |
| 22 | TSPYL5 | 829.00 | 72.33 | 3.10 | $1.19 \times 10^{-9}$ | 57 | AMOTL1 | 2535.84 | 68.85 | 3.67 | $3.94 \times 10^{-6}$ |
| 23 | HERC5 | 11068.07 | 2191.72 | 1.45 | $1.98 \times 10^{-9}$ | 58 | MANEAL | 354.94 | 982.06 | −1.69 | $4.72 \times 10^{-6}$ |
| 24 | UBE2QL1 | 205.44 | 53.77 | 3.16 | $2.23 \times 10^{-9}$ | 59 | SPATS2L | 7329.68 | 2990.72 | 0.97 | $4.97 \times 10^{-6}$ |
| 25 | ARHGAP6 | 3915.43 | 352.19 | 3.53 | $2.73 \times 10^{-9}$ | 60 | VEGFB | 6472.65 | 5415.88 | 0.90 | $5.21 \times 10^{-6}$ |
| 26 | SLAIN1 | 6757.48 | 3157.85 | 1.24 | $2.73 \times 10^{-9}$ | 61 | ATP1B1 | 7552.46 | 859.00 | 2.47 | $5.25 \times 10^{-6}$ |
| 27 | CERS6 | 5027.25 | 5425.85 | −1.13 | $3.74 \times 10^{-9}$ | 62 | SIX3 | 800.36 | 1203.65 | −6.36 | $5.25 \times 10^{-6}$ |
| 28 | ATP8B1 | 296.62 | 13.02 | 3.73 | $5.99 \times 10^{-9}$ | 63 | LOC285972 | 1639.86 | 2658.04 | −1.16 | $7.04 \times 10^{-6}$ |
| 29 | GRIA3 | 43.74 | 504.25 | −3.95 | $7.66 \times 10^{-9}$ | 64 | MYO18A | 8284.31 | 9316.46 | −0.69 | $8.77 \times 10^{-6}$ |
| 30 | MARCH8 | 1078.97 | 1225.73 | −1.64 | $7.68 \times 10^{-9}$ | 65 | L1TD1 | 67.03 | 1.01 | 8.23 | $8.90 \times 10^{-6}$ |
| 31 | DUSP4 | 17734.90 | 5898.88 | 1.94 | $1.58 \times 10^{-8}$ | 66 | RRP7B | 3521.07 | 2614.81 | 0.94 | $9.80 \times 10^{-6}$ |
| 32 | EPB41L5 | 1929.07 | 494.79 | 1.93 | $1.70 \times 10^{-8}$ | 67 | SPARC | 4705.62 | 16484.18 | −1.60 | $1.51 \times 10^{-5}$ |
| 33 | ZNF711 | 1265.24 | 3592.15 | −3.34 | $1.05 \times 10^{-7}$ | 68 | ESF1 | 32558.55 | 26226.19 | 0.69 | $1.60 \times 10^{-5}$ |
| 34 | RGS2 | 1264.73 | 83.46 | 3.85 | $1.26 \times 10^{-7}$ | 69 | FUT8 | 10906.46 | 16945.75 | −0.94 | $1.64 \times 10^{-5}$ |
| 35 | TP53BP2 | 2231.99 | 622.32 | 2.13 | $1.41 \times 10^{-7}$ | 70 | MIR363 | 109.28 | 0.00 | 8.71 | $1.71 \times 10^{-5}$ |

A list of the 70 most significantly deregulated mRNAs in *FTSJ1* LCL mutants versus controls.

**Table 4.  Bibliographic search on miRNAs deregulated in *FTSJ1* loss of function LCL mutant cell lines.**

| miRNA | Brain-related | Brain cancer–related | Cancer-related |
|---|---|---|---|
| hsa-miR-20b-5p | — | — | Khuu et al (2016) |
| hsa-miR-222-3p | Lau et al (2013), Kretschmann et al (2015), Kan et al (2012), Risbud & Porter (2013) | Gillies & Lorimer (2007), Zhang et al (2010) | — |
| hsa-miR-548ax | — | Neuroblastoma for other miR-548 family members | Watahiki et al (2011) (also others cancers for other miR-548 family members) |
| hsa-miR-125b-2-3p | yes | yes | yes |
| hsa-miR-221-3p | Kretschmann et al (2015), Kan et al (2012), Risbud & Porter (2013), Ding et al (2016), Ma et al (2016), Roser et al (2018) | (see miR-222-3p) | Fornari et al (2008) |
| hsa-miR-335-3p | yes | yes | yes |
| hsa-miR-181b-2-3p | (see miR(181a-5p)) | (see miR(181a-5p)) | (see miR(181a-5p)) |
| hsa-miR-99a-5p | yes | yes | yes |
| hsa-miR-10a-5p | Gui et al (2015), Roser et al (2018) | Tehler et al (2011), Lund (2010) | Tehler et al (2011), Lund (2010) |
| hsa-miR-181b-3p | (see miR(181a-5p)) | (see miR(181a-5p)) | (see miR(181a-5p)) |
| hsa-miR-106a-5p | yes | yes | yes |
| hsa-miR-181a-2-3p | (see miR(181a-5p)) | (see miR(181a-5p)) | (see miR(181a-5p)) |
| hsa-miR-146a-5p | yes | yes | yes |
| hsa-miR-4482-3p | — | — | — |
| hsa-miR-125b-5p | yes | yes | yes |
| hsa-miR-450b-5p | — | — | yes |
| hsa-miR-424-3p | yes | — | yes |
| hsa-miR-363-3p | Lau et al (2013), Kiyosawa et al (2019) | Conti et al (2016), Qiao et al (2013) | Jiang et al (2018), Ye et al (2017), Hu et al (2016), Wang et al (2016), Karatas et al (2016), Chapman et al (2015), Zhang et al (2016), Khuu et al (2016) |
| hsa-let-7c-5p | — | — | — |
| hsa-miR-450a-5p | yes | — | yes |
| hsa-miR-18b-5p | — | — | — |
| hsa-miR-550a-3p | — | — | yes |
| hsa-miR-181a-5p | Zhang et al (2017), Ding et al (2016), Roser et al (2018) | Shi et al (2008) | Yang et al (2017), Li et al (2015) |
| hsa-miR-550b-2-5p | — | — | yes |
| hsa-miR-181a-3p | (see miR(181a-5p)) | (see miR(181a-5p)) | (see miR(181a-5p)) |
| hsa-miR-181b-5p | (see miR(181a-5p)) | (see miR(181a-5p)) | (see miR(181a-5p)) |
| hsa-miR-183-5p | yes | yes | yes |
| hsa-miR-99a-3p | yes | yes | yes |
| hsa-miR-135a-5p | yes | yes | yes |
| hsa-miR-146b-5p | yes | yes | yes |
| hsa-miR-542-5p | — | yes | yes |
| hsa-miR-944 | yes | — | yes |
| hsa-miR-625-5p | — | — | — |
| hsa-miR-625-3p | — | — | — |
| hsa-miR-4772-5p | — | — | yes |
| hsa-miR-182-5p | yes | yes | yes |
| Total # | 24 | 23 | 30 |

The list shows for each miRNA whether any link was found to brain development or brain-related diseases, also to cancer and specifically to brain cancers. The references are given for most of the miRNAs. The colour code of the miRNA names indicates whether they were found to be up- (red) or down-regulated (blue) in *FTSJ1* mutant LCLs derived from XLID affected individuals compared with control LCLs derived from healthy individuals.

# Materials and Methods

### FTSJ1 variants and LCLs

The various LCLs were generated using established methods from blood samples of XLID affected or healthy male individuals. The cells were cultured in RPMI 1640 medium with L-glutamine and sodium bicarbonate (ref. R8758-500ML; Sigma–Aldrich) supplemented with 10% FBS (Gibco) and 1% penicillin–streptomycin (ref. P0781; Sigma-Aldrich) at 37 °C with 5% $CO_2$. Cells were split at ½ dilution ~24 h before being collected for RNA extraction with TRI Reagent (Sigma-Aldrich) following the manufacturer's instructions.

6514AW and 6514JW (LCL65AW and LCL65JW in this study): Family A3—LCLs from two brothers with mild or severe ID associated with psychiatric manifestations (anger, aggression, anxiety, depression, schizophrenia requiring medication) bearing a splice variant in *FTSJ1:* c.121 + 1delG (Freude et al, 2004). This variant leads to a retention of intron 2, creating a premature stop codon (p.Gly41Valfs*10). Part of the transcripts undergo nonsense-mediated mRNA decay.

11716IJ (LCL11 in this study): Family A18—LCL from one male with moderate to severe ID without dysmorphic features carrying an interstitial microdeletion at Xp11.23. The extent of the deletion was subsequently delineated to about 50 kb by regular PCR and included only the *SLC38A5* and *FTSJ1* genes. qPCR with the *FTSJ1*-ex3 primers is negative, thus demonstrating the complete deletion of the *FTSJ1* locus (Froyen et al, 2007).

22341SR (LCL22 in this study): Family 7 (A26P)—LCL from one male with moderate ID and psychiatric features (mild anxiety and compulsive behaviour) carrying a missense mutation c.76G > C; p.Ala26Pro in *FTSJ1*. This family has been reported previously (Guy et al, 2015).

LCL-MM: This is a newly reported family. The LCL has been generated from one male with mild ID, facial dysmorphia (hypertelorism, pointed chin, ears turned back), speech delay, attention disorders, and behavioural problems carrying a hemizygous de novo variant c.362-2A > T in *FTSJ1*. The mutation is predicted to disrupt the acceptor splice site of exon 6 (NM_012280.3: c.362-2A > T). This variant causes a skipping of the entire exon 6 in the mRNA (r.362_414del) leading to a frameshift and a premature stop codon (p.Val121Glyfs*51) (Fig S1A). Part of the transcripts undergo nonsense-mediated mRNA decay (Fig S1C). Consequently, a strong decrease of the corresponding mRNA steady-state level is observed (Fig S1B). This variant was deposited in the *ClinVar* database (VCV000981372.1). The research on LCL-MM was performed according to a research protocol approved by a local ethics committee (Comité Consultatif de Protection des Personnes dans la Recherche Biomédicale—CCPPRB). Written informed consent was obtained from the patient and his legal representatives.

18451PK (LCL18 in this study), 16806JD (LCL16 in this study), 3-2591 (LCL25 in this study), and 3-5456 (LCL54 in this study): LCL established from control males. Four LCLs not mutated in the *FTSJ1* gene from unaffected males of a similar age were used as controls. Written informed consent was obtained from those individuals, and previously described LCLs were obtained from patients and their legal representatives in the original publications described above.

### LCL MM variant characterization at the mRNA level

As the *FTSJ1* mRNA was highly down-regulated in LCL MM, characterization of the *FTSJ1* transcript for this experiment was performed on total RNAs from cells treated with cycloheximide (see the NMD inhibition test section protocol below). This allowed a threefold increase in *FTSJ1* mRNA in LCL MM (Fig S1B). 1 µg of total RNAs from wild-type LCL 25 and LCL MM was treated with DNase I (M0303S; New England Biolabs), and RT was carried out with random hexamer primers (S0142; Thermo Fisher Scientific) using SuperScript III Reverse Transcriptase (18080-044; Invitrogen), following the supplier's protocol. *FTSJ1* cDNAs were amplified from 2 µl of RT reaction using the following PCR primers: Forward: 5'-GGCAGTTGACCTGTGTGCAGC-3'; Reverse: 5'-CCCTCTAGGTCCAGTGGGTAAC-3'. PCR products were sequenced using the Sanger method with a forward primer hybridizing in exon 5: 5'-CCACTGCCAAGGAGATCA-3' (Fig S1A). Sequences are available upon request. Briefly, this variant causes a skipping of the entire exon 6 in the mRNA leading to a frameshift and a premature stop codon, thus undergoing nonsense-mediated mRNA decay as shown in Fig S1C. Consequently, a strong decrease of the corresponding mRNA steady-state level is observed (Fig S1B). This MM variant was deposited in the *ClinVar* database (VCV000981372.1).

### RiboMethSeq

RiboMethSeq analysis of human LCL tRNAs was performed as described in Marchand et al (2017). Briefly, tRNAs extracted from LCLs were fragmented in 50 mM bicarbonate buffer, pH 9.2, for 15 min at 95°C. The reaction was stopped by ethanol precipitation. The pellet was washed with 80% ethanol, and sizes of generated RNA fragments were assessed by capillary electrophoresis using a small RNA chip on Bioanalyzer 2100 (Agilent Technologies). RNA fragments were directly 3'-end–dephosphorylated using 5 U of Antarctic phosphatase (New England Biolabs) for 30 min at 37°C. After inactivation of the phosphatase for 5 min at 70°C, RNA fragments were phosphorylated at the 5'-end using T4 PNK and 1 mM ATP for 1 h at 37°C. End-repaired RNA fragments were then purified using RNeasy MinElute Cleanup Kit (QIAGEN) according to the manufacturer's recommendations. RNA fragments were converted to library using NEBNext Small RNA Library Kit (ref. E7330S; New England Biolabs) following the manufacturer's instructions. DNA library quality was assessed using a High Sensitivity DNA chip on Bioanalyzer 2100. Library sequencing was performed on Illumina HiSeq 1000 in a single-read mode for 50 nt. Primary analysis of sequencing quality was performed with RTA 2.12 software, to ensure > Q30 quality score for >95% of obtained sequences.

After SR50 sequencing run, demultiplexing was performed with BclToFastq v2.4, and reads not passing quality filter were removed. Raw reads after demultiplexing were trimmed with Trimmomatic v0.32 (Bolger et al, 2014). Alignment to the reference tDNA sequences was performed with bowtie 2 ver2.2.4 (Langmead et al, 2009) in end-to-end mode. Uniquely mapped reads were extracted from *.sam file by RNA ID and converted to *.bed format using bedtools v2.25.0 (Quinlan, 2014). Positional counting of 5'- and 3'-ends of each read was performed with awk Unix command. Further treatment steps were performed in R environment (v3.0.1). In brief, 5'- and 3'-end counts were merged together by RNA position and used for the calculation of ScoreMEAN (derived from MAX Score)

(Pichot et al, 2020), and Scores A and B (Birkedal et al, 2015) and MethScore (Score C) (Marchand et al, 2016). Scores were calculated in the window of –2 to +2 neighbouring nucleotides. Profiles of RNA cleavage at selected (candidate and previously known) positions were extracted and visually inspected.

Analysis of human tRNA 2′-O-methylation by RiboMethSeq was performed using the optimized non-redundant collection of reference tRNA sequences. This reduced collection contains 43 tRNA species and was validated by analysis of several experimentally obtained RiboMethSeq datasets (Pichot et al, 2021). Alignment of RiboMethSeq reads obtained in this study also confirmed low content in ambiguously mapped reads. In order to establish a reliable map of Nm positions in the human tRNA anticodon loop, RiboMethSeq cleavage profiles were used to calculate detection scores (Mean and ScoreA2) (Pichot et al, 2020). However, this scoring strategy shows its limits in the case of short and highly structured RNAs (like tRNAs), because the cleavage profile is highly irregular. In addition, because these scores are calculated for two neighbouring nucleotides, the simultaneous loss of two closely located Nm residues (e.g., $Cm_{32}$ and $Gm_{34}$ in $tRNA^{Phe}$) makes analysis of raw score misleading (Angelova et al, 2020). Moreover, the presence of multiple RT-arresting modifications (Anreiter et al, 2021) in the same tRNAs ($m^1A$, $m^1G$, $m^2_2G$, $m^3C$, etc.) reduces coverage in the upstream regions. Considering all these limitations, visual inspection of raw cleavage profiles revealed to be the most appropriate, because changes in protection of a given nucleotide represent modulation of its Nm methylation status. Analysis of alignment statistics demonstrated that most of the human tRNAs are well represented in the analysed datasets and proportion of uniquely mapped reads were >90% for all tRNA sequences, except the tRNA Leu_CCA family, composed of three highly similar species. Only limited coverage of totally mapped reads <7,500 reads/tRNA (~100 reads/position) was obtained for five tRNAs (Arg_TCG, Leu_CAA2, Ser_CGA_TGA1, Thr_CGT, and Tyr_ATA).

To identify potential Nm32/Nm34 residues, raw cleavage profiles of the 11-nt region around pos 33 were visually inspected and profiles for WT samples were compared with *FTSJ1* mutants. Because of the limited number of mapped raw reads, coverage in the anticodon loop for Leu_CAA, Ser_CGA_TGA1, Thr_CGT, and Tyr_ATA was insufficient; thus, these species were excluded from further analysis. The results of this analysis are given in Table 2. This analysis allowed to identify 10 Nm32 and four Nm34 modifications on the tRNA ACL. Inosine residues formed by deamination of A34 at the wobble tRNA position (FTSJ1-independent) are visible in the sequencing data and are also shown in Table 2. 10 Nm32 and three Nm34 modifications were found to be FTSJ1-dependent. The only exception is Cm34 in tRNAMet_CAT known to be formed by snoRNA-guided fibrillarin (Vitali & Kiss, 2019). Comparison of these data with previously reported Nm modifications in the human tRNA anticodon loop demonstrated that 2/3 of the observed sites have been described, either in tRNAdb2009 ([Jühling et al, 2009], http://trnadb.bioinf.uni-leipzig.de/) or in two recent studies that used LC–MS/MS analysis (Li et al, 2020; Nagayoshi et al, 2021). Table 2 also shows those modifications in other organisms including yeast, mice, and *Drosophila*. We were not able to confirm Nm residues previously reported in tRNASec_TCA (Nm34) and tRNAVal_AAC(Cm32); however, because of sequence similarity, tRNAVal_AAC clusters together with two other tRNAVal (CAC and TAC1). tRNALeu_AAG and Leu_TAG have similar sequences and thus were not distinguished by sequencing; however, Nm32 was detected.

## mRNA sequencing and data analysis

mRNA sequencing was performed as in Khalil et al (2018). 5 µg of total RNA was treated with 1 MBU of DNase (BaseLine-Zero DNase; Epicentre) for 20 min at 37°C to remove residual genomic DNA contamination. RNA quality was verified by a PicoRNA chip on Bioanalyzer 2100 (Agilent Technologies) to ensure RIN (RNA integrity number) > 8.0. PolyA + fraction was isolated from 4.5 µg of DNase-treated total RNA using NEBNext Oligo d(T)25 Magnetic Beads Kit (New England Biolabs), according to the manufacturer's recommendations. PolyA + enrichment and the absence of residual ribosomal RNA contamination were verified using PicoRNA chips on Bioanalyzer 2100 (Agilent Technologies). PolyA + fraction (1 ng for each sample) was used for whole-transcriptome library preparation using ScriptSeq v2 RNA-seq Kit (Illumina). Libraries amplified in 14 PCR cycles were purified using Agencourt AMPure XP Beads (Beckman Coulter), at a ratio 0.9× to remove adapter dimer contamination. Quality of the libraries was verified by HS DNA Chip on Bioanalyzer 2100 (Agilent Technologies) and quantification done by Qubit 2.0 with an appropriate RNA quantification kit. Sequencing was performed on HiSeq 1000 (Illumina) in single-read SR50 mode. About 50 million of raw sequencing reads was obtained for each sample. Adapters were trimmed by Trimmomatic v0.32 (Bolger et al, 2014) and the resulting sequencing reads aligned in sensitive-local mode by Bowtie 2 v2.2.4 (Langmead & Salzberg, 2012) to hg19 build of human genome. Differential expression was analysed using *.bam files in the DESeq2 package (Love et al, 2014) under R environment. Analysis of KEGG and gene ontology pathways for differentially expressed genes was done under R environment.

## Small RNA sequencing and data analysis

Small RNA-seq libraries were generated from 1,000 ng of total RNA using TruSeq Small RNA Library Prep Kit (Illumina), according to the manufacturer's instructions. Briefly, in the first step, RNA adapters were sequentially ligated to each end of the RNA, first the 3′ RNA adapter that is specifically modified to target microRNAs and other small RNAs, then the 5′ RNA adapter. Small RNA ligated with 3′ and 5′ adapters was reverse-transcribed and PCR-amplified (30 s at 98°C; [10 s at 98°C, 30 s at 60°C, 15 s at 72°C] × 13 cycles; 10 min at 72°C) to create cDNA constructs. Amplified cDNA constructs of 20–40 nt were selectively isolated by acrylamide gel purification followed by ethanol precipitation. The final cDNA libraries were checked for quality and quantified using capillary electrophoresis and sequenced on the Illumina HiSeq 4000 at the Institut de Génétique et de Biologie Moléculaire et Cellulaire (IGBMC) GenomEast sequencing platform.

For small RNA data analysis, adapters were trimmed from total reads using FASTX_Toolkit (http://hannonlab.cshl.edu/fastx_toolkit/). Only trimmed reads with a length between 15 and 40 nucleotides were kept for further analysis. Data analysis was performed according to the published pipeline ncPRO-seq (Chen et al, 2012). Briefly, reads were mapped onto the human genome assembly hg19 with Bowtie v1.0.0. The annotations for miRNAs were done

with miRBase v21. The normalization and comparisons of interest were performed using the test for differential expression, proposed by Love et al (2014) and implemented in the Bioconductor package DESeq2 v1.22.2 (http://bioconductor.org/). MicroRNA target prediction was performed using miRNet 2.0 (Chang et al, 2020).

## Northern blotting

For Northern blotting analysis of tRNAs, 5 μg of total RNA from human LCLs was resolved on 15% urea–polyacrylamide gels for ~2 h in 0.5× TBE buffer at 150 V, then transferred to Hybond-NX membrane (GE Healthcare) in 0.5× TBE buffer for 1 h at 350 mA of current and EDC–cross-linked for 45 min at 60°C with a solution containing 33 mg/ml of 1-ethyl-3-(3-dimethylaminopropyl) carbodiimide (EDC) (Sigma-Aldrich), 11 ng/μl of 1-methylimidazole, and 0.46% of HCl. The membranes were first prehybridized for 1 h at 42°C in a hybridization buffer containing 5×SSC, 7% SDS, 5.6 mM NaH$_2$PO$_4$, 14.4 mM Na$_2$HPO$_4$, and 1× Denhardt's solution. DNA oligonucleotide probes were labelled with $^{32}$P at the 5'-end by T4 polynucleotide kinase following the manufacturer's instructions (Fermentas). The membranes were hybridized with the labelled probes overnight at 42°C in the hybridization buffer, then washed twice for 15 min in wash buffer A (3× SSC and 5% SDS) and twice in wash buffer B (1× SSC and 1% SDS) before film exposure at −80°C for variable time durations. Probe sequences are available in the Primers, probes, and sequences section.

## qRT–PCR

RNA was extracted from human LCLs using TRI Reagent (Sigma-Aldrich). After DNase digestion of total RNA using the TURBO DNA-free Kit (Ambion), 1 μg was used in a RT reaction with random primers (Promega) and RevertAid Reverse Transcriptase (ref. EP0442; Thermo Fisher Scientific). The cDNA was used to perform qPCR on a CFX96 Touch Real-Time PCR Detection System (Bio-Rad) using target-specific primers. *hGAPDH* was used for normalization (Primers, probes, and sequences section). The analysis was performed using ΔΔ Ct, on at least three biological replicates. Statistical analysis using a bilateral *t* test was performed, and *P*-values were calculated.

## NMD inhibition test

LCLs were seeded in 25-cm cell culture flasks at a density of 3 × 10$^6$ cells and treated with 100 μg/ml of cycloheximide or equal volume of water as a control for 6 h. Cells were harvested by centrifugation (75003607 Rotor; Thermo Fisher Scientific) at 1,000 rpm for 5 min and flash-frozen in liquid nitrogen. RNA extraction was carried out using TRI Reagent (Sigma-Aldrich) following the supplier's protocol. DNase I digestion was carried out using RNase-free DNase I (M0303S; New England Biolabs), and RT on 1 μg of DNase-treated total RNA was performed using RevertAid Reverse Transcriptase. Quantitative PCR was performed as specified above using specific primers for *FTSJ1* and *GAPDH*.

## miRNA complementation experiments

mirVana miRNA mimics and inhibitors were used for *hsa-miR-181a-5p* overexpression/inhibition (4464066 and 4464084; Ambion). HeLa cells were transfected with corresponding mirVana miRNA in 24-well plates at a density of 20,000 cells per well, using Lipofectamine RNAiMAX (Cat #13778100; Invitrogen). We set up the transfection ratios to 15 pmol of miRNA mimic/μl of Lipofectamine, and 30 pmol of miRNA inhibitor/μl of Lipofectamine. Cells were harvested 48 h post-transfection and assayed for target gene expression. miRNA quantification was performed by qRT–PCR on *miR-181a-5p* using QIAGEN's miRCURY LNA miRNA PCR System. RT is performed using miRCURY LNA RT Kit (339340) and qPCR using miRCURY LNA SYBR Green PCR Kit (339346). LNA-enhanced primers were used for miRNA SYBR Green qPCR (refer to the list of primers and probes).

## Primers, probes, and sequences

Northern blot analysis was performed using *hsa-miR-181a-5p*–specific probes with the following sequences: 5'-AACATTCAACGCTGTCGGT-GAGT-3' (sense probe) and 5'-ACTCACCGACAGCGTTGAATGTT-3' (antisense probe). Human U6–specific probe was used for detecting U6 as a loading control: 5'-GCAAGGATGACACGCAAATTCGTGA-3' (sense probe) and 5'-TCACGAATTTGCGTGTCATCCTTGC-3' (antisense probe). qPCR analysis (after an RT reaction performed with random primers) was performed with the use of primers with the following sequences:

| Target gene | Primer | Sequence |
|---|---|---|
| BTBD3 | Forward | 5'-TGGCAGATGTACATTTTGTGG-3' |
| | Reverse | 5'-AACACAGAGCTCCCAACAGC-3' |
| SPARC | Forward | 5'-GAGAAGGTGTGCAGCAATGA-3' |
| | Reverse | 5'-AAGTGGCAGGAAGAGTCGAA-3' |
| GAPDH | Forward | 5'-CAACGGATTTGGTCGTATTGG-3' |
| | Reverse | 5'-GCAACAATATCCACTTTACCAGAGTTAA-3' |
| FTSJ1 | Forward | 5'-CCATTCTTACGACCCAGATTTCA-3' |
| | Reverse | 5'-CCCTCTAGGTCCAGTGGGTAAC-3' |
| ZNF711 | Forward | 5'-CACACGCCAGACTCTAGAATGG-3' |
| | Reverse | 5'-CCATTCCAGCCACAAAATCTTG-3' |
| hsa-miR-181a-5p | Cat #339306; QIAGEN | GeneGlobe ID—YP00206081 |
| UniSp6 (miRNA Spike in) | Cat #339306; QIAGEN | GeneGlobe ID—YP00203954 |

UCG isodecoder sequences (refer to Table 2) >hs_tRNA Arg_CCG_TCG_(UCG1)_gaccgcgtggcctaatggataaggcgtctgacttcggatcagaagattgagggttcgagtcccttcgtggtcgcca > hs_tRNAArg_TCG_(UCG2) _ggccgngtggcctaatggataaggcgtctgacttcggatcanaagattgcaggttngagtn ctgccncggtcgcca.

## iPSC culture and maintenance

iPSC cell line WTSIi002 purchased from EBISC (European bank for induced pluripotent cells) was maintained on feeder-free conditions on Geltrex LDEV-Free hESC-qualified Reduced Growth Factor Basement Membrane Matrix (A1413302; Thermo Fisher Scientific) in Essential 8 Flex Media Kit (A2858501; Thermo Fisher Scientific) with 0,1% penicillin–streptomycin (15140122; Thermo Fisher Scientific).

## iPSC differentiation in dorsal NPCs

To obtain neural progenitor cells (NPCs) from the dorsal telencephalon, embryoid bodies (EBs) were formed by incubating iPSC clusters with Accutase (A1110501; Thermo Fisher Scientific) for 7 min at 37°C and dissociated into single cells. To obtain EB of the same size, $3 \times 10^6$ cells were added per well in the AggreWell 800 plate (34815; STEMCELL Technologies) with Essential 8 Flex Media supplemented with Stemgent hES Cell Cloning & Recovery Supplement (1×, STE01-0014-500; Ozyme) and incubated at 37°C with 5% $CO_2$ (day 1). After 24 h in culture (day 0), EBs from each microwell were collected by pipetting up and down the medium several times and transferred into Corning non-treated culture dishes (CLS430591-500EA; Merck) in EB medium containing DMEM/F12 GlutaMAX (35050061; Thermo Fisher Scientific), 20% KnockOut Serum Replacement (10828028; Thermo Fisher Scientific), 1% non-essential amino acid (11140035; Thermo Fisher Scientific), 0,1% penicillin–streptomycin (15140122; Thermo Fisher Scientific), and 100 $\mu$M 2-mercaptoethanol (31350010; Thermo Fisher Scientific), supplemented with two inhibitors of the SMAD signalling pathway, 2.5 $\mu$M dorsomorphin (P5499; Sigma-Aldrich) and 10 $\mu$M SB-431542 (ab120163; Abcam). EB medium supplemented as described previously was changed every day for 5 d. On day 6, floating EBs are plated on 0.01% poly-L-ornithine (P4957; Sigma-Aldrich)– and 5 $\mu$g/ml laminin (L2020; Sigma-Aldrich)-coated dishes for rosette expansion in Neurobasal minus vitamin A (10888; Thermo Fisher Scientific), B-27 supplement without vitamin A (12587; Thermo Fisher Scientific), 1% GlutaMAX (35050061; Thermo Fisher Scientific), 0,1% penicillin–streptomycin (15140122; Thermo Fisher Scientific), and 100 $\mu$M 2-mercaptoethanol (31350010; Thermo Fisher Scientific). The neural medium was supplemented with 10 ng/ml epidermal growth factor (AF-100-15; PeproTech) and 10 ng/ml basic fibroblast growth factor (234-FSE-025; R&D Systems). From days 6 to 10, the medium was changed every day until the appearance of rosettes. On day 10, rosettes are manually picked up using a syringe and dissociated with Accutase, then seeded on poly-L-ornithine/laminin-coated dishes for expansion of dorsal NPCs. They were maintained with passage for two additional weeks to achieve a large pool of neural precursor cells (NPCs).

## NPC drug treatment

NPCs are seeded in poly-L-ornithine– and laminin-coated coverslips in 24-well plates at a density of $2 \times 10^5$ cells per well. After 48 h, the medium is changed and combined with 100 $\mu$M of 2,6-diaminopurine (DAP) (247847; Sigma-Aldrich) or equal volume of sterile $H_2O$.

## NPC immunostainings

24 h after DAP treatment, NPCs were fixed in 4% paraformaldehyde for 10 min, permeabilized, and blocked for 45 min with blocking buffer (PBS supplemented with 0.3% Triton X-100, 2% horse serum). Primary antibodies, Sox2 (1/500, AB5603; Millipore) and DCX (1/2,000, AB2253; Millipore), were incubated overnight at 4°C using the same solution. Cells were rinsed three times with PBS and incubated for 1 h at RT with secondary antibodies and DAPI (1/10,000, D9564; Sigma-Aldrich) diluted in the same solution and rinsed three times with PBS before mounting on slides with VectaShield Vibrance mounting medium.

## Neuronal cell image acquisitions

Images were acquired in z-stacks using a confocal microscope Nikon A1R HD25 with a 60× objective. Images were flattened with a max-intensity Z-projection.

## Neurogenesis quantification

All cells (DAPI) from each acquisition were numbered using Fiji's point tool. Cells expressing DCX (immature neurons) and SOX2 (NPCs and intermediates, which also started expressing DCX) were also numbered on five to six microscopy images. Over 1,400 cells were numbered for each condition in triplicate. A ratio of DCX-expressing cells is calculated over the total cell number and expressed in fold change and compared between DAP-treated and untreated cells.

## Branching quantifications

All DCX-expressing neurons were traced using Simple Neurite Tracer from the Neuroanatomy plugin by Fiji. Length measurements of traces were performed using the Simple Neurite Tracer Measure Menu, and thin projections were counted manually using Fiji's point tool. Quantifications were performed on five acquisitions, and each IF experiment was done in triplicate. Ratios for the number of thin projections/neuron length (mm) were calculated and compared between DAP-treated and control cells.

## *Drosophila* NMJ analysis

For NMJ staining, third instar larvae were dissected in cold PBS and fixed with 4% paraformaldehyde in PBS for 45 min. Larvae were then washed in PBST (PBS + 0.5% Triton X-100) six times for 30 min and incubated overnight at 4°C with mouse anti-synaptotagmin, 1:200 (3H2 2D7, Developmental Studies Hybridoma Bank, DSHB). After six 30-min washes with PBST, secondary antibody anti-mouse conjugated to Alexa Fluor 488 and TRITC-conjugated anti-HRP (Jackson

ImmunoResearch) were used at a concentration of 1:1,000 and incubated at room temperature for 2 h. Larvae were washed again six times with PBST and finally mounted in Vectashield (Vector Laboratories).

For DAP treatment, freshly hatched *Canton-S* flies were collected and placed on a normal food medium containing 600 $\mu$M of 2,6-diaminopurine (DAP) ( 247847; Sigma-Aldrich). After 5 d, third instar larvae were dissected and subjected to NMJ staining.

Images from muscles 6–7 (segments A2–A3) were acquired with a Zeiss LSM 710 confocal microscope. Serial optical sections at 1,024 × 1,024 pixels with 0.4 $\mu$m thickness were obtained with the ×40 objective. Bouton number was quantified using Imaris 9 software. ImageJ software was used to measure the muscle area and the NMJ axon length and branching. Statistical tests were performed in GraphPad (Prism 8).

### *Drosophila* behavioural assays

Flies were raised at 25°C for associative memory assays and the corresponding controls. All behavioural experiments were performed on young adults (1–3 d-old). All behavioural experiments were performed on starved flies, which is a prerequisite for appetitive conditioning with a sucrose reinforcement. 0–2 d after hatching, flies were put on starvation for 21 h at 25°C on mineral water (Evian).

### Appetitive memory assay

Appetitive associative conditioning was performed in custom-designed barrel-type apparatus as previously described (Colomb et al, 2009), which allows the parallel conditioning of three groups of flies. The odorants 3-octanol and 4-methylcyclohexanol, diluted in paraffin oil at a final concentration of 0.29 g·l⁻¹, were used for conditioning and for the test of memory retrieval. Groups of 20–50 flies were subjected to one cycle of appetitive olfactory conditioning as follows: throughout the conditioning protocol, flies were submitted to a constant airflow at 0.6 litres·min⁻¹. After 90 s of habituation, flies were first exposed to an odorant (the CS⁺) for 1 min, whereas given access to dried sucrose, flies were then exposed 45 s later to a second odorant without shocks (the CS⁻) for 1 min. 3-Octanol and 4-methylcyclohexanol were alternately used as CS⁺ and CS⁻. The memory test was performed in a T-maze apparatus. Each of the two arms of the T-maze was connected to a bottle containing one odorant (either 3-octanol or 4-methylcyclohexanol) diluted in paraffin oil. The global airflow from both arms of the T-maze was set to 0.8 litres·min⁻¹. Flies were given 1 min in complete darkness to freely move within the T-maze. Then, flies from each arm were collected and counted. The repartition of flies was used to calculate a memory score as $(N_{CS+} - N_{CS-})/(N_{CS+} + N_{CS-})$. A single performance index value is the average of two scores obtained from two groups of genotypically identical flies conditioned in two reciprocal experiments, using either odorant as the CS⁺. Thus, values of performance index range between –1 and +1, with the value of 0 (equal repartition) corresponding to "no memory." The indicated "n" is the number of independent performance index values for each genotype. LTM performance was assessed 24 h (±2 h) after conditioning, and short-term memory, 1 h (±30 min) after conditioning.

### *Innate odour avoidance and sucrose attraction assay*

Innate sucrose preference was measured in a T-maze. Flies were given the choice for 1 min between one arm of the T-maze coated with dried sucrose, and one empty arm. There was no airflow in the T-maze for this assay. Flies were then collected from each arm and counted; an attraction index was calculated as $(N_{sucrose} - N_{empty})/(N_{sucrose} + N_{empty})$. The side of the T-maze with sucrose was alternated between experimental replicates. Innate odour avoidance was measured in a T-maze. One arm of the T-maze was connected to a bottle containing the tested odorant (3-octanol or 4-methylcyclohexanol) diluted in paraffin oil, and the other arm was connected to a bottle containing paraffin oil only. The global airflow from both arms of the T-maze was set to 0.8 litres·min⁻¹. Flies were given 1 min in complete darkness to freely move within the T-maze. Flies were then collected from each arm and counted; an avoidance index was calculated as $(N_{air} - N_{odour})/(N_{air} + N_{odour})$. The side of the T-maze with odorant-interlaced air was alternated between experimental replicates.

### *Quantification and statistical analysis*

All data are presented as the mean ± SEM. Performances from different groups (mutant and control) were statistically compared using one-way ANOVA followed by Tukey's post hoc pairwise comparison between the mutant genotypes and the control group.

## Data Availability

The RNA-sequencing and small RNA-sequencing data discussed in this publication are deposited and fully accessible, either in NCBI's Gene Expression Omnibus accessible through GEO Series accession number GSE179384 for small RNA-seq or at the European Nucleotide Archive at EMBL-EBI under accession number PRJEB46400 for the RiboMethSeq and PRJEB46399 for RNA-seq.

## Supplementary Information

## Acknowledgements

We thank the patients and their families for their participation in the study. We also thank Christelle Thibaut-Charpentier from the GenomEast sequencing platform in Strasbourg, a member of the "France Génomique" consortium (ANR-10-INBS-0009), the Institut de Génétique Médicale d'Alsace, for their technical support, and Myriam Bronner from Nancy University Hospital for the establishment of the LCL-MM line. We thank Johann Schor and Laura Guédon for their help with behavioural experiments. Human NPC work was carried out at ICM's CELIS core facility and NPC imaging at the ICM Quant facility. We thank Dr. Bernard Moss (NIAID/NIH) for providing *FTSJ1* KO HeLa cells. We thank the members of the TErBio laboratory for helpful discussions and reading of the article. C Carré received financial support from the CNRS, Sorbonne Université (Emergence 2021_RNA-Mod-Diag), the Fondation Maladies Rares (Genomics-2018 #11809 and Genomics-2020 #12824), the IBPS-2020 Action Incitative, the Ligue National contre le cancer Île de France (RS21/76-29), and ANR (ANR-21-CE12-0022-01 #BiopiC). Work in the BA Hassan laboratory was supported by the Investissements d'Avenir programme

(ANR-10-IAIHU-06), Paris Brain Institute-ICM core funding, the Roger De Spoelberch Foundation Prize, and a grant from the Neuro-Glia Foundation. Research in the laboratory of J-Y Roignant is supported by University of Lausanne and the Deutsche Forschungsgemeinschaft RO 4681/9-1, RO4681/12-1, and RO4681/13-1. DG Dimitrova and M Brazane have PhD fellowships from the Ministère de la Recherche et de l'Enseignement Supérieur at the doctoral school Complexité du Vivant (ED515). We also thank the Fondation ARC pour la Recherche sur le Cancer and the FRM (Fondation pour la Recherche Médicale) for funding support to DG Dimitrova, MT Angelova, and M Brazane (FDT202204014985), fourth-year PhD; "Réseau André Picard," the "Société Française de Génétique" (to M Brazane, MT Angelova, and DG Dimitrova); and COST action "EPITRAN" CA16120 (to Y Motorin, J-Y Roignant, V Marchand, DG Dimitrova, M Brazane, MT Angelova, and C Carré) for travelling and training fellowships.

## Author Contributions

M Brazane: data curation, formal analysis, validation, investigation, visualization, methodology, and writing—original draft, review, and editing.

DG Dimitrova: data curation, formal analysis, validation, investigation, visualization, methodology, and writing—original draft, review, and editing.

J Pigeon: resources, formal analysis, validation, methodology, and writing—review and editing.

C Paolantoni: data curation, formal analysis, validation, investigation, and visualization.

T Ye: data curation, software, formal analysis, validation, visualization, methodology, and writing—review and editing.

V Marchand: data curation, software, validation, and methodology.

B Da Silva: data curation, validation, investigation, visualization, and methodology.

E Schaefer: resources and writing—review and editing.

MT Angelova: formal analysis, validation, investigation, methodology, and writing—review and editing.

Z Stark: resources and writing—review and editing.

M Delatycki: resources and writing—review and editing.

T Dudding-Byth: resources and writing—review and editing.

J Gecz: resources, formal analysis, validation, methodology, and writing—review and editing.

P-Y Plaçais: resources, formal analysis, validation, investigation, visualization, methodology, and writing—review and editing.

L Teysset: validation, investigation, methodology, and writing—review and editing.

T Préat: formal analysis, validation, investigation, visualization, methodology, and writing—review and editing.

A Piton: resources, data curation, formal analysis, methodology, and writing—review and editing.

BA Hassan: resources, methodology, and writing—review and editing.

J-Y Roignant: formal analysis, validation, investigation, visualization, methodology, and writing—original draft, review, and editing.

Y Motorin: data curation, software, formal analysis, validation, investigation, visualization, methodology, and writing—original draft, review, and editing.

C Carré: conceptualization, resources, data curation, software, formal analysis, supervision, funding acquisition, validation, investigation, visualization, methodology, project administration, and writing—original draft, review, and editing.

## Conflict of Interest Statement

The authors declare that they have no conflict of interest.

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
