## [Reviewer comments · Life Science Alliance]

Life Science Alliance

The ribose methylation enzyme FTSJ1 has a conserved role in neuron morphology & learning performance

Mira Brazane, Dilyana Dimitrova, Julien Pigeon, Chiara Paolantoni, Tao Ye, Virginie Marchand, Bruno Da Silva, Elise Schaefer, Margarita Angelova, Zornitza Stark, Martin Delatycki, Tracy Dudding - Byth, Jozef Gecz, Pierre-Yves Placais, Laure Teyssset, Thomas Preat, Amélie Piton, Bassem Hassan, Jean-Yves Roignant, Yuri Motorin, and Clément Carré

DOI: <https://doi.org/10.26508/lsa.202201877>

Corresponding author(s): Clément Carré, Sorbonne University, Centre National de la Recherche Scientifique, Laboratoire de Biologie du Développement - Institut de Biologie Paris Seine

Review Timeline:

Submission Date:	2022-12-15
Editorial Decision:	2023-01-04
Revision Received:	2023-01-08
Accepted:	2023-01-10

Transaction Report:

Please note that the manuscript was reviewed at Review Commons and these reports were taken into account in the decision-making process at Life Science Alliance.

We thank the reviewers for their comments on our manuscript. To address their criticisms we have prepared a point by point response as shown below in red.

Reviewer #1 (Evidence, reproducibility and clarity (Required)):

Dimitrova et al performed RiboMethSeq on tRNAs from human LCLs carrying control or FTSJ1 pathogenic variant to demonstrate the full spectrum of human FTSJ1 tRNA substrates. And the level of several mRNAs and miRNAs seemed to be deregulated in affected human FTSJ1 individuals LCLs. In addition, the authors uncovered that the defect of FTSJ1 lead to the morphological abnormality in human neural progenitor cells and Drosophila. Moreover, in Drosophila, the defect of Trm7_34 resulted in long-term memory deficit.

Loss of tRNA modifications are closely associated with kinds of human diseases, highlighting the crucial roles of tRNA modification. Thus, the study on the relationship between tRNA modification and human disease are very important. This work adds light on the FTSJ1 pathologies in human intellectual disability, and contains the research value. However, there are several issues need to be addressed:

We thank the Reviewer#1 for recognizing the importance of our work.

****Major comments:****

1) As shown in Table 2, through RiboMethSeq, the identified tRNAs of human FTSJ1 are tRNAArg(UCG), tRNAGln(CUG) and tRNAGly(GCC) at position 32, tRNALeu(CAG) at position 34, tRNAPhe(GAA) and tRNATrp(CCA) at position 32 and 34. Compared with that of human FTSJ1 in Li et al. 2020, Nagayoshi et al. 2021, and Kawarada et al. in 2017, the tRNA substrates in this work are quite limited. I guess whether the alkaline hydrolysis conditions are too harsh, thus causing degradation of tRNAs? The authors need to check the quality of their RNA fragments library first.

We thank the reviewer for this insightful comment. As explained below, the RiboMethSeq has some limitations, which may lead to the identification of false positives. To prevent this issue we decided to make a conservative call. In this revised version we make clearer the detected positions in Table 2 and describe in more detail the methods and analysis of RiboMethSeq data in the Material and Methods. Finally, the text below is now included in the presented revised manuscript for a better understanding of these RiboMethSeq detection analysis, detection reports and potential limitations.

Analysis of human tRNA 2'-O-methylation by RiboMethSeq (Marchand et al. 2017) was performed as described previously (Birkedal et al. 2015; Marchand et al. 2016), using the optimized non-redundant collection of reference tRNA sequences. This reduced collection contains 43 tRNA species and was validated by analysis of several experimentally obtained RiboMethSeq sequencing datasets (Pichot et al. 2021).

Alignment of RiboMethSeq reads obtained in this study also confirmed low content in ambiguously mapped reads. In order to establish a reliable map of Nm positions in human tRNA anticodon loop, RiboMethSeq cleavage profiles were used to calculate detection scores (Mean and ScoreA2) (Pichot et al. 2020). However, this scoring strategy shows its

limits in the case of short and highly structured RNAs (like tRNAs), since the cleavage profile is highly irregular. In addition, since these scores are calculated for 2 neighboring nucleotides, simultaneous loss of two closely located Nm residues (e.g. Cm32 and Gm34 in tRNAPhe) makes analysis of raw score misleading (Angelova et al. 2020). Moreover, the presence of multiple RT-arresting modifications (Anreiter et al. 2021) in the same tRNAs (m1A, m1G, m22G, m3C, etc) reduces coverage in the upstream regions. Considering all these limitations, visual inspection of raw cleavage profiles revealed to be the most appropriate, since changes in protection of a given nucleotide represents modulation of its Nm methylation status.

Analysis of alignment statistics demonstrated that the majority of human tRNAs are well represented in the analysed datasets and proportion of uniquely mapped reads were >90% for all tRNA sequences, except tRNA^{Leu}(CCA) family, composed of 3 highly similar species. Only limited coverage of totally mapped reads <7500 reads/tRNA (~100 reads/position) was obtained for 5 tRNAs (Arg_TCG, Leu_CAA2, Ser_CGA_TGA1, Thr_CGT and Tyr_ATA).

In order to identify potential Nm32/Nm34 residues, raw cleavage profiles of the 11 nt region around pos 33 were visually inspected and profiles for WT samples were compared to FTSJ1 mutants. Due to the limited number of mapped raw reads, coverage in the anticodon loop for Leu_CAA, Ser_CGA_TGA1, Thr_CGT and Tyr_ATA was insufficient; thus, these species were excluded from further analysis. The results of this analysis are given in Table 2. This analysis allowed to identify 14 tRNA species bearing altogether 12 Nm32 and 4 Nm34 residues. Inosine residues formed by deamination of A34 at the wobble tRNA position are visible in the sequencing data and are also shown. Ten Nm32 and 3 Nm34 modifications were found to be FTSJ1-dependent. The only exception is Cm34 in tRNA^{Met}_CAT known to be formed by snoRNA-guided Fibrilarin (Vitali and Kiss 2019).

Comparison of these data with previously reported Nm modifications in human tRNA anticodon loop demonstrated that 2/3 of the observed sites have been described, either in tRNAdb2009 (Jühling et al. 2009, <http://trnadb.bioinf.uni-leipzig.de/>), or in two recent studies used LC-MS/MS analysis (Nagayoshi et al. 2021; Li et al. 2020). Table 2 also shows those modifications in other organisms including yeast, mouse and Drosophila. We were not able to confirm Nm residues previously reported in tRNA^{Sec}_TCA (Nm34) and tRNA^{Val}_AAC(Cm32), however, due to sequence similarity, tRNA^{Val}_AAC clusters together with two other tRNA^{Val} (CAC and TAC1). tRNA^{Leu}_AAG and Leu_TAG have similar sequences and thus were not distinguished by sequencing, however Nm32 was detected.

2) The authors found the mRNAs steady state level and miRNA populations were deregulated in the affected human FTSJ1 LCLs. In these cells, the mRNA of ZNF711 and SPARC were decreased, and the mRNA of BTBD3 were increased (figure 2B). miR-181a-5p level was upregulated (figure 3C). These are exciting discoveries, and it would be better to test more mRNAs and miRNAs. And the wild-type FTSJ1 needs to be complemented to verify whether the deregulation of mRNA and miRNA could be restored.

We thank the reviewer for finding our discovery exciting. Other miRNAs and mRNAs are available in the mRNAseq and smallRNAseq data set presented in Table 3 and 4 and

Figures S2A and S2B. All miRNAs and mRNAs expression checked by qPCR (and northern blot for miR-181a-5p) were consistent with the original mRNAseq and smallRNAseq data.

Regarding the complementation, we attempted to complement FTSJ1 mutant cells using a FTSJ1_Flag expression vector (Nagayoshi et al. 2021). LCLs are known to be extremely difficult to transfect (poor transfection rate), thus, we attempted this complementation experiment using HeLa cells mutated for FTSJ1 (Sivan et al. 2018). Unfortunately, the transfection of CMV_FTSJ1_Flag gives rise to a low steady state level of FTSJ1 mRNA in FTSJ1 KO cells when compared to the control. This was confirmed by several repeated experiments as well as western blot that shows a good expression of FTSJ1_Flag in control cells and no detection in FTSJ1 KO HeLa cells. We have repeated these experiments several times and always face this “expression” problem in FTSJ1 KO cells. We will be pleased to provide the corresponding experiments (RT-qPCR and western blot) to the reviewers if needed.

3) It seems credible that the SPARC mRNA downregulation may correlated with the upregulated miR-10a-5p, and the upregulation of miR-181a-5p may lead to the upregulation of BTBD3. But this hypothesis needs to be verified through complementing the sense or anti-sense miRNAs to detect whether the deregulation of the target genes could be restored.

We have performed these experiments (KD or OE of *miR-181a-5p*) as suggested by the Reviewer. This was performed in HeLa cells for the same reason as explained above (LCL are hardly if not transfectable). We included these new experiments in a new supplemental Figure (Figure S4) and corresponding results section as well as discussed these new results in the discussion section as depicted below:

Results section:

This cross-analysis revealed that the BTBD3 gene is potentially targeted by miR-181a-5p (He et al, 2015), the two of which were upregulated in NSXLID affected individuals-derived LCLs (Figures 3A, 3C and Table 4), implicating a possible connection between them that differs from the canonical miRNA silencing pathway. LCL are known to be hardly transfectable, however miR-181a-5p and BTBD3 are expressed similarly in HeLa cells (Figure S4A). Thus, by mimicking miR-181a-5p expression or repression, we show that miR-181a-5p silences BTBD3 in HeLa cells (Figure S4B), strongly suggesting that BTBD3 mRNA is a bona fide target of miR-181a-5p. Strikingly, in FTSJ1 mutant cells, the silencing activity of miR-181a-5p on BTBD3 is compromised in both HeLa and LCL.

Discussion section:

In addition to the reported prediction (He et al. 2015), we show that BTBD3 is a bona fide miR-181a-5p target. Surprisingly, both BTBD3 and miR-181a-5p were up-regulated in FTSJ1 depleted patient cells. While the precise mechanism is not known yet our results suggest that Nm-MTases genes could act upstream of small RNA biogenesis and function through transcriptional downregulation of Argonaute mRNA in Drosophila FTSJ1 mutants (Angelova and Dimitrova et al. 2020) and in human cells (not shown). On the other hand, tRNA fragments (tRF) abundance seen in FTSJ1 mutant fly (Angelova and Dimitrova et al. 2020) and mice (Nagayoshi et al. 2021) can associate with Dicer, Argonaute and Piwi proteins, thus affecting their silencing function. Such tRF-mediated titration of proteins away from

canonical substrates has been previously reported in Drosophila and human cell lines (Durdevic et al. 2013; Goodarzi et al. 2015).

Additionally, more miRNAs and the targeted mRNAs should be tested to support the conclusion that FTSJ1 deregulated miRNAs target FTSJ1 deregulated mRNAs. It would be better to declare "FTSJ1 deregulated miRNAs may correlate with FTSJ1 deregulated mRNAs" in the title of this part.

We now make our conclusion clearer in the light of our new experiments as well as tune down the title as proposed. The fact that we have to use HeLa cells limits the number of miRNA::mRNA potential couples that are similarly expressed in LCLs. We tested 4 couples, and only 1 was expressed similarly in HeLa and LCL: *BTBD3* and *miR-181a-5p*. Unfortunately, both *SPARC* and *miR-10a-5p* are poorly expressed in HeLa cells, preventing us to KD or OE *miR-10a-5p* and looking at the impact on *SPARC* mRNA. However, this *SPARC::miR-10a-5p* couple was already confirmed experimentally (Bryant et al, 2012; Wang et al, 2020), thus strongly suggesting that *SPARC* mRNA downregulation observed in *FTSJ1* mutants LCL may be due to its increased silencing by the upregulated *miR-10a-5p*.

4) The abnormal neuron morphology was observed in human neural progenitor cells which were treated with inhibitor of FTSJ1, as well in Drosophila, and the long-term memory was impacted in Drosophila with *Trm7_34* mutant. Considering that the mRNAs and miRNAs were both deregulated in the affected human FTSJ1 LCLs, I wonder whether there is the relationship between the changes of mRNAs and the above neuron phenotypes. It may be helpful to test the mRNA levels of some neuron morphology and long-term memory related genes.

In addition to the already identified *BTBD3* mRNA which is deregulated in FTSJ1 mutant cells (RNAseq and qPCR), we tested *ROBO1*, a gene involved in neurogenesis. Unfortunately, *ROBO1* was poorly expressed in LCL and no significant change was detected by mRNAseq, nor RTqPCR. Despite the blood origin of LCL, *AHNAK* (FC: -2.62817811520946; padj: 7.59976433793026e-05), a neuroblast differentiation-associated protein and *BTBD3* were both deregulated in FTSJ1 mutated cells (seen in mRNAseq analysis Table 3 and European Nucleotide Archive (ENA) accession number PRJEB46399 and RTqPCR). Importantly *BTBD3* activity is known to direct the dendritic field orientation during development of the sensory neuron in mice cortex (Matsui et al, 2013) and to regulate mice behaviors (Thompson et al, 2019). We found that *BTBD3* mRNA was significantly upregulated in both mRNA-seq and RT-qPCR analyses (Figure 2B), which is in accordance with the phenotype observed in NPC inhibited for FTSJ1 activity (Figure 4).

5) I wonder whether the authors conducted the appetitive conditioning assay on Drosophila with *Trm7_32* mutant, or both *Trm7_32* and *Trm7_34* mutant, respectively. These experiments on the two mutants are conducive to a better understanding of the biological functions of 2'-O-methylation at positions 32 and 34.

We thank the reviewer for this suggestion. Initially only the *Trm7_34* mutant was tested. Now we include 3 new genotypes: *Trm7_32*, *Trm7_34* and *Trm7_32, Trm7_34* double mutant on appetitive conditioning assay was performed as requested by the three reviewers. It is now depicted in a new Figure 6 and clearly shows that both positions on tRNA are important for

LTM (and not STM). We discuss these new results in the discussion section as depicted below:

A synaptic overgrowth was also observed in Drosophila, indicating that this function of FTSJ1 is conserved across evolution. In addition we found that the long term memory but not the short term was significantly altered in the absence of FTSJ1 in flies. This is consistent with the learning deficits observed in mice and humans. In contrast to Human FTSJ1 and the yeast ortholog TRM7, Drosophila uses two distinct paralogs to methylate positions 32 and 34, respectively, on tRNAs ACL. Interestingly, we found that the lack of both, Trm7_34 and Trm7_32 had an effect on long term memory, suggesting that the methylation at wobble position 34 and 32 are critical for this function. However, the lack of both modifications (as in mammals Ftsj1 mutant) is not cumulative regarding the memory deficit (Figure. 6). This last observation is strongly supported by the affected human individual that harbours a missense variant (p.Ala26Pro, LCL22 in this study), resulting in loss of Gm₃₄, but not of Cm₃₂ in human tRNA^{Phe} (Guy et al, 2015). Further studies should aim to understand how the loss of methylation at these ACL positions affects the learning and memory functions.

6) The discoveries (if make the mechanism clear) in this manuscript could be organized to a fascinating story, however, the current version is more like to pile the data without a theme, necessary links are missed in the current version. For example, how the loss of tRNA modifications leads to mRNA/miRNA deregulation? Or how the mRNA/miRNA deregulation cause the abnormal neuron morphology? A further study on any of the links could make the research profound and significant.

We acknowledge the comment of the Reviewer and we revised our manuscript accordingly to make those theme links clearer. Regarding the miRNAs affected in FTSJ1 mutated cells, we hypothesized that the translation/stability of key miRNA pathway proteins could be involved in the observed miRNA deregulation (as we suggested earlier in *Angelova et al., NAR 2020*). Concerning the mRNA deregulation, in our initial submission we also pointed out that some mRNAs deregulated in FTSJ1 mutated cells are targeted by miRNA (*i.e. SPARC mRNA is an experimentally confirmed target of mir-10a-5p...*).

In this revised version, we now show that *BTBD3* gene is regulated by *miR-181a-5p*. *BTBD3* activity is known to direct the dendritic field orientation during development of the sensory neuron in mice cortex and to regulate mice behaviours. The discussion section is now extended to better describe the potential link of the deregulated miRNA and mRNA in FTSJ1 mutated cells as depicted below:

Our transcriptomic analysis also highlighted novel transcripts and miRNA targets that may play important roles in the development of the diseases. For instance we found 36 differentially expressed miRNAs, most of which were already associated with brain diseases and functioning and/or cancer development. Strikingly, the most prevalent associated cancer types were the ones related to the brain tissues. Consistently with the post-transcription regulation role of miRNA, we also found through mRNA-seq an enrichment of brain morphogenesis-related mRNAs differentially expressed in FTSJ1 loss of function when compared to control LCLs. Interestingly, a cross-analysis of these two RNA sequencing experiments revealed potential miRNA::target mRNA couples among the deregulated RNA

populations. This is indicative of possible miRNA silencing changes in the absence of FTSJ1, similarly to what we report earlier in *Drosophila* FTSJ1 mutant orthologs. The predicted miRNA::mRNA couples need to be further validated individually in neuronal tissues, although their report from miRnet database (<https://www.mirnet.ca/>) already includes experimental evidence on the miRNA::mRNA regulation, particularly for BTBD3 and SPARC mRNAs (Bryant et al, 2012; Wang et al, 2020; He et al, 2015). We show here that BTBD3 is a bona fide miR-181a-5p target. Surprisingly, both BTBD3 and miR-181a-5p were up-regulated in FTSJ1 depleted patient cells. While the precise mechanism is not known yet our results suggest that Nm-MTases genes could act upstream of small RNA biogenesis and function through transcriptional downregulation of Argonaute mRNA in *Drosophila* FTSJ1 mutants (Angelova and Dimitrova et al. 2020) and in human cells (not shown). On the other hand, tRNA fragments (tRF) abundance seen in FTSJ1 mutant fly (Angelova and Dimitrova et al. 2020) and mice (Nagayoshi et al. 2021) can associate with Dicer, Argonaute and Piwi proteins, thus affecting their silencing function. Such tRF-mediated titration of proteins away from canonical substrates has been previously reported in *Drosophila* and human cell lines (Durdevic et al. 2013; Goodarzi et al. 2015).

****Minor comments:****

1) Nagayoshi et al. 2021 has revealed some tRNA substrates of human and mouse FTSJ1. The authors need to cite this reference in the introduction and text.

We fully agree, in fact we previously cited Nagayoshi et al. 2021 in the main text. As suggested we now cite this reference in the introduction, as well as in Table 2 and in the discussion section.

2) According to the MS studies in Li et al. 2020, tRNA^{Arg}(ACG), -(CCG) and -(UCG) of human FTSJ1 only contains Nm at position 32. The MS studies of Table 2 should be revised. And, the authors should cite the related references of the identified tRNAs of human, *Drosophila* and *S. cerevisiae* Trm7 in the annotation of Table 2.

The table 2 now provides more information including references for Human tRNA detected in this study and previous ones. In addition, we now provide more information on other organisms for the same positions (mice, flies and yeast).

3) In figure 6, the order of pictures can be placed as the described in the manuscript to facilitate readers' reading.

We modified new Figure 6 with the new data accordingly to facilitate readers' reading.

Reviewer #1 (Significance (Required)):

Loss of tRNA modifications are closely associated with kinds of human diseases, highlighting the crucial roles of tRNA modification. Thus, the study on the relationship between tRNA modification and human disease are very important. This work adds light on the FTSJ1 pathologies in human intellectual disability, and contains the research value. The discoveries (if make the mechanism clear) in this manuscript could be organized to a

fascinating story, however, the current version is more like to pile the data without a theme, necessary links are missed in the current version. For example, how the loss of tRNA modifications leads to mRNA/miRNA deregulation? Or how the mRNA/miRNA deregulation cause the abnormal neuron morphology? A further study on any of the links could make the research profound and significant.

We thank and understand the comments of this Reviewer and we hope that the revised manuscript and new experiments detailed above will be satisfying.

Reviewer #2 (Evidence, reproducibility and clarity (Required)):

This is a well conducted study into the biological function of the ribose methylation enzyme FTSJ1 using human patient derived cell lines and a *Drosophila* model. They map the Nm modifications in tRNAs by analysing patient samples and detect a number of differentially expressed genes and miRNAs that are relevant to the pathology. Since patients suffer a number of neurological disorders including intellectual disability, they analyse neurite formation in human differentiated cells using an inhibitor of FTSJ1 to find differences in neurite branching. Compellingly, they can recapitulate this phenotype in a *Drosophila* model in the double mutant and using an inhibitor. Then they go on to show that *Trm7_34* mutants have a long-term memory phenotype.

We thank the Reviewer for his comments and for pointing out that our study is well conducted.

****Major comments:****

Since the branching phenotype at *Drosophila* NMJs is only seen in the double mutants, I am surprised that the role of tRNA Nm methylation in the learning phenotype was not analysed more comprehensively. This analysis of *Trm7_34* and the double mutant needs to be added for a complete picture, particularly since the flies are viable and available.

This proposed experiment, *Trm7_32* or/ and *Trm7_32* and *Trm7_34* double mutant on appetitive conditioning assay was performed as requested by the three reviewers. It is now depicted in a new Figure 6 and clearly shows that both positions on tRNA are important for LTM (and not STM). We discuss these new results in the discussion section as depicted below:

*A synaptic overgrowth was also observed in *Drosophila*, indicating that this function of FTSJ1 is conserved across evolution. In addition we found that the long term memory but not the short term was significantly altered in the absence of FTSJ1 in flies. This is consistent with the learning deficits observed in mice and humans. In contrast to Human FTSJ1 and the yeast ortholog TRM7, *Drosophila* uses two distinct paralogs to methylate positions 32 and 34, respectively, on tRNAs ACL. Interestingly, we found that the lack of both, *Trm7_34* and *Trm7_32* had an effect on long term memory, suggesting that the methylation at wobble position 34 and 32 are critical for this function. However, the lack of both modifications (as in mammals *Ftsj1* mutant) is not cumulative regarding the memory deficit (Figure. 6). This last observation is strongly supported by the affected human*

individual that harbours a missense variant (p.Ala26Pro, LCL22 in this study), resulting in loss of Gm₃₄, but not of Cm₃₂ in human tRNA^{Phe} (Guy et al, 2015). Further studies should aim to understand how the loss of methylation at these ACL positions affects the learning and memory functions.

P1: Since Nm is prominently present in cap-adjacent nucleotides, this should be mentioned in the introduction in more detail also referring to CMTr enzymes to make the distinction to FTSJ1 clear.

We agree with the Reviewer. This point has been added in the introduction section as suggested.

P12 bottom/p18 top: Cm32 tRNAArg(UCG) has been detected with MS. There is substantial discrepancy between the detection by Nm-Seq and MS in Table 1 and they only add an explanation for one (which should be marked * to indicate presence in raw reads) why it is detected with MS and not Nm-seq. This issue should be picked up in the Discussion by referencing to a general difficulty in mapping modifications in RNA e.g. by citing PMID: 32620324

We agree with the Reviewer and now we provide a Table 2 easier to read in our revised Manuscript, including references of Nm detection in 4 different organisms. In addition, as suggested by Reviewer1 and 2, we now provide explanations for this discrepancy in the Material and Methods and Discussion sections including citing the suggested reference. Finally, the text below is now included in the presented revised manuscript for a better understanding of these RiboMethSeq detection analysis, detection reports and potential limitations.

Analysis of human tRNA 2'-O-methylation by RiboMethSeq (Marchand et al. 2017) was performed as described previously (Birkedal et al. 2015; Marchand et al. 2016), using the optimized non-redundant collection of reference tRNA sequences. This reduced collection contains 43 tRNA species and was validated by analysis of several experimentally obtained RiboMethSeq sequencing datasets (Pichot et al. 2021).

Alignment of RiboMethSeq reads obtained in this study also confirmed low content in ambiguously mapped reads. In order to establish a reliable map of Nm positions in human tRNA anticodon loop, RiboMethSeq cleavage profiles were used to calculate detection scores (Mean and ScoreA2) (Pichot et al. 2020). However, this scoring strategy shows its limits in the case of short and highly structured RNAs (like tRNAs), since the cleavage profile is highly irregular. In addition, since these scores are calculated for 2 neighboring nucleotides, simultaneous loss of two closely located Nm residues (e.g. Cm32 and Gm34 in tRNA^{Phe}) makes analysis of raw score misleading (Angelova et al. 2020). Moreover, the presence of multiple RT-arresting modifications (Anreiter et al. 2021) in the same tRNAs (m1A, m1G, m22G, m3C, etc) reduces coverage in the upstream regions. Considering all these limitations, visual inspection of raw cleavage profiles revealed to be the most appropriate, since changes in protection of a given nucleotide represents modulation of its Nm methylation status.

Analysis of alignment statistics demonstrated that the majority of human tRNAs are well represented in the analysed datasets and proportion of uniquely mapped reads were >90% for all tRNA sequences, except tRNA^{Leu}(CCA) family, composed of 3 highly similar species. Only limited coverage of totally mapped reads <7500 reads/tRNA (~100 reads/position) was obtained for 5 tRNAs (Arg_TCG, Leu_CAA2, Ser_CGA_TGA1, Thr_CGT and Tyr_ATA).

In order to identify potential Nm32/Nm34 residues, raw cleavage profiles of the 11 nt region around pos 33 were visually inspected and profiles for WT samples were compared to FTSJ1 mutants. Due to the limited number of mapped raw reads, coverage in the anticodon loop for Leu_CAA, Ser_CGA_TGA1, Thr_CGT and Tyr_ATA was insufficient; thus, these species were excluded from further analysis. The results of this analysis are given in Table 2. This analysis allowed to identify 14 tRNA species bearing altogether 12 Nm32 and 4 Nm34 residues. Inosine residues formed by deamination of A34 at the wobble tRNA position are visible in the sequencing data and are also shown. Ten Nm32 and 3 Nm34 modifications were found to be FTSJ1-dependent. The only exception is Cm34 in tRNA^{Met}_CAT known to be formed by snoRNA-guided Fibrilarin (Vitali and Kiss 2019).

Comparison of these data with previously reported Nm modifications in human tRNA anticodon loop demonstrated that 2/3 of the observed sites have been described, either in tRNAdb2009 (Jühling et al. 2009, <http://trnadb.bioinf.uni-leipzig.de/>), or in two recent studies used LC-MS/MS analysis (Nagayoshi et al. 2021; Li et al. 2020). Table 2 also shows those modifications in other organisms including yeast, mouse and Drosophila. We were not able to confirm Nm residues previously reported in tRNA^{Sec}_TCA (Nm34) and tRNA^{Val}_AAC(Cm32), however, due to sequence similarity, tRNA^{Val}_AAC clusters together with two other tRNA^{Val} (CAC and TAC1). tRNA^{Leu}_AAG and Leu_TAG have similar sequences and thus were not distinguished by sequencing, however Nm32 was detected.

P12/p18: Although the focus here is on tRNA modification by FTSJ1, why a number of miRNAs and mRNAs are differentially expressed remains enigmatic, but is highly relevant to understand the pathology. At least a paragraph detailing how this could come about would be instructive, e.g. are there indications from the differentially expressed genes?

As suggested and in light of our new experiments on miRNA::mRNA couple, we now added two paragraphs and new results (Figure S4) to make this point clearer.

Results section:

This cross-analysis revealed that the BTBD3 gene is potentially targeted by miR-181a-5p (He et al, 2015), the two of which were upregulated in NSXLID affected individuals-derived LCLs (Figures 3A, 3C and Table 4), implicating a possible connection between them that differs from the canonical miRNA silencing pathway. LCL are known to be hardly transfectable, however miR-181a-5p and BTBD3 are expressed similarly in HeLa cells (Figure S4A). Thus, by mimicking miR-181a-5p expression or repression, we show that miR-181a-5p silences BTBD3 in HeLa cells (Figure S4B), strongly suggesting that BTBD3 mRNA is a bona fide target of miR-181a-5p. Strikingly, in FTSJ1 mutant cells, the silencing activity of miR-181a-5p on BTBD3 is compromised in both HeLa and LCL.

Discussion section:

In addition to the reported prediction (He et al. 2015), we show that BTBD3 is a bona fide miR-181a-5p target. Surprisingly, both BTBD3 and miR-181a-5p were up-regulated in FTSJ1

depleted patient cells. While the precise mechanism is not known yet our results suggest that Nm-MTases genes could act upstream of small RNA biogenesis and function through transcriptional downregulation of Argonaute mRNA in Drosophila FTSJ1 mutants (Angelova and Dimitrova et al. 2020) and in human cells (not shown). On the other hand, tRNA fragments (tRF) abundance seen in FTSJ1 mutant fly (Angelova and Dimitrova et al. 2020) and mice (Nagayoshi et al. 2021) can associate with Dicer, Argonaute and Piwi proteins, thus affecting their silencing function. Such tRF-mediated titration of proteins away from canonical substrates has been previously reported in Drosophila and human cell lines (Durdevic et al. 2013; Goodarzi et al. 2015).

Also, did the authors check for Nm in mRNAs and miRNAs and are there differences in the differentially expressed genes? If they can't detect Nm in mRNA and miRNA because it is below the threshold, or they can conclude that there are no changes, they should mention/discuss this accordingly.

We thank the Reviewer for these important comments. Up to now, no clear data demonstrates clearly that FTSJ1 Nm MTase family of enzymes is able to modify other RNA than tRNA (and possibly rRNA). In addition, to date, detecting Nm on low abundant RNA such as mRNA and miRNA is notoriously challenging and falls below the detection by RiboMethSeq as discussed in (Anreiter et al. 2021).

P15 top: connection?

The connection discussed in p15 top (mutant of *Drosophila* FTSJ1 and small non-coding RNA failure) is now discussed in more detail in the revised manuscript, particularly in the discussion section as depicted below:

While the precise mechanism is not known yet our results suggest that Nm-MTases genes could act upstream of small RNA biogenesis and function through transcriptional downregulation of Argonaute mRNA in Drosophila FTSJ1 mutants (Angelova and Dimitrova et al. 2020) and in human cells (not shown). On the other hand, tRNA fragments (tRF) abundance seen in FTSJ1 mutant fly (Angelova and Dimitrova et al. 2020) and mice (Nagayoshi et al. 2021) can associate with Dicer, Argonaute and Piwi proteins, thus affecting their silencing function. Such tRF-mediated titration of proteins away from canonical substrates has been previously reported in Drosophila and human cell lines (Durdevic et al. 2013; Goodarzi et al. 2015).

P18 top Cm32 tRNAArg(UCU) is not shown in table 1, why? The table and referencing in the text of the different tRNAs need to be correct!

We apologize if this point was not sufficiently clear. We now provide a clearer Table 2 and corresponding citing references. Regarding Cm32 tRNAArg(UCU), this modification (Cm) is not detected nor described elsewhere to our knowledge. Indeed tRNAArg(UCU) is m³C modified not Nm. This is the reason why we do not show tRNAArg(UCU) in Table 2 (Table 2:

Nm modification deposited by FTSJ1). We discuss this point in detail in the discussion section as depicted below:

In the case of tRNA^{Arg(UCG)}, we confirmed not only a new target for FTSJ1, but also a modification which was not previously reported in modomics but only recently in HEK293 FTSJ1 CRISPR mutant (Li et al, 2020). Indeed C₃₂ was known to be m³C and not Nm modified for the two other isoacceptors (tRNA^{Arg(CCU)} and tRNA^{Arg(UCU)}) (Boccaletto et al, 2018).

Reviewer #2 (Significance (Required)):

These results are exciting and will path the way to a more mechanistic analysis on how exactly tRNA modifications impact on differential expression of genes that result in a neurological phenotype.

We thank the Reviewer for the careful reading, reviewing and suggestions as well as finding our results exciting.

Reviewer #3 (Evidence, reproducibility and clarity (Required)):

In this manuscript the authors investigate several aspects of mutations in FTSJ1, previously linked to non-syndromic X-linked intellectual disability (NSXLID), associated with loss of ribose 2'-O-methylation at C32 and particularly at N34 in substrate tRNAs. The authors report a novel FTSJ1 pathogenic variant that results in skipping of exon 6, leading to a frame shift, premature termination, and reduced levels of FTSJ1 mRNA due to NMD. RiboMethSeq analysis was then used for unbiased analysis of 2'-O-methylation levels of FTSJ1 tRNA substrates, and results showed that lymphoblastoid cell lines (LCL) from this patient had undetectable levels of Cm32 and Nm34 of several known substrate tRNAs, as also found by parallel analysis of each of 4 LCLs derived from NSXLID patients with other FTSJ1 mutations. In addition, this RiboMethSeq analysis identified two new FTSJ1 target tRNAs (C32 of tRNAGln(CUG) and tRNAGly(CCC)), and did not confirm three other reported sites. , and did not confirm some other reported sites attributed to FTSJ1 (C32 of tRNAArg(ACG), tRNAArg(CCG), and tRNAGln(UUG), and C34 of tRNAArg(CCG)). Subsequent transcriptome analysis revealed 686 dysregulated genes in FTSJ1 mutant LCLs, compared to WT, with significant enrichment for GO terms related to brain morphogenesis, and enrichment for terms related to mitochondrial biology. Analysis of miRNAs revealed 36 dysregulated miRNAs that were statistically significant, with significant overlap of miRNAs identified in other brain diseases and in brain-related cancers, and with at least two miRNAs possibly regulating dysregulated mRNAs. In addition, treatment of neural progenitor cells with the FTSJ1 inhibitor diaminopurine (DAP) resulted in increased interstitial protrusion on neurites, consistent with significant synaptic overgrowth in the corresponding Drosophila trm732 trm734 double mutants, and Drosophila larvae treated with DAP. Finally, the authors provide evidence that long term memory is impaired in Drosophila mutants lacking trm7_34.

Overall this manuscript has several excellent and intriguing results, but in the opinion of this reviewer, many of them are not as well fleshed out as one would like. As written, the manuscript is a disparate collection of interesting data on a number of aspects related to FTSJ1, but no aspect seems complete.

We thank the Reviewer for the careful reading, reviewing and suggestions as well as finding our results *excellent*. We believe that the addition of the new proposed experiments as well as an improved presentation of some results now make the study more complete and clearer.

1.

a. The RiboMethSeq results are not completely presented. As written, two new FTSJ1 sites are found, but the underlying data -- a figure showing the reads at each residue in the anticodon loop -- should be included for these new sites. Similarly, 3 well-documented previously reported FTSJ1 modification sites (ref. 32) are not found in the RiboMethSeq analysis, but there is no data other than the summary in Table 2, and no way for the reader to judge if the lack of detection of these reported sites is convincing.

We apologize for the unclear presented data and now provide them in the revised version particularly in new Table 2 and Figure 1, Figure S1D as well as providing the raw data set of RiboMethSeq at the European Nucleotide Archive (ENA) at EMBL-EBI under accession number PRJEB46400. In addition we included a section in the Material and Methods and core text on the benefits and limitations of using RiboMethSeq for Nm detection in comparison to MS, and why the identified targets can, in some specific known situations, partially overlap (see below).

Analysis of human tRNA 2'-O-methylation by RiboMethSeq (Marchand et al. 2017) was performed as described previously (Birkedal et al. 2015; Marchand et al. 2016), using the optimized non-redundant collection of reference tRNA sequences. This reduced collection contains 43 tRNA species and was validated by analysis of several experimentally obtained RiboMethSeq sequencing datasets (Pichot et al. 2021).

Alignment of RiboMethSeq reads obtained in this study also confirmed low content in ambiguously mapped reads. In order to establish a reliable map of Nm positions in human tRNA anticodon loop, RiboMethSeq cleavage profiles were used to calculate detection scores (Mean and ScoreA2) (Pichot et al. 2020). However, this scoring strategy shows its limits in the case of short and highly structured RNAs (like tRNAs), since the cleavage profile is highly irregular. In addition, since these scores are calculated for 2 neighboring nucleotides, simultaneous loss of two closely located Nm residues (e.g. Cm32 and Gm34 in tRNAPhe) makes analysis of raw score misleading (Angelova et al. 2020). Moreover, the presence of multiple RT-arresting modifications (Anreiter et al. 2021) in the same tRNAs (m1A, m1G, m22G, m3C, etc) reduces coverage in the upstream regions. Considering all these limitations, visual inspection of raw cleavage profiles revealed to be the most appropriate, since changes in protection of a given nucleotide represents modulation of its Nm methylation status.

Analysis of alignment statistics demonstrated that the majority of human tRNAs are well represented in the analysed datasets and proportion of uniquely mapped reads were >90% for all tRNA sequences, except tRNALeu(CCA) family, composed of 3 highly similar species. Only limited coverage of totally mapped reads <7500 reads/tRNA (~100 reads/position) was obtained for 5 tRNAs (Arg_TCG, Leu_CAA2, Ser_CGA_TGA1, Thr_CGT and Tyr_ATA).

In order to identify potential Nm32/Nm34 residues, raw cleavage profiles of the 11 nt region around pos 33 were visually inspected and profiles for WT samples were compared to FTSJ1 mutants. Due to the limited number of mapped raw reads, coverage in the anticodon loop for Leu_CAA, Ser_CGA_TGA1, Thr_CGT and Tyr_ATA was insufficient; thus, these species were excluded from further analysis. The results of this analysis are given in Table 2. This analysis allowed to identify 14 tRNA species bearing altogether 12 Nm32 and 4 Nm34 residues. Inosine residues formed by deamination of A34 at the wobble tRNA position are visible in the sequencing data and are also shown. Ten Nm32 and 3 Nm34 modifications were found to be FTSJ1-dependent. The only exception is Cm34 in tRNAMet_CAT known to be formed by snoRNA-guided Fibrilarin (Vitali and Kiss 2019).

Comparison of these data with previously reported Nm modifications in human tRNA anticodon loop demonstrated that 2/3 of the observed sites have been described, either in tRNADB2009 (Jühling et al. 2009), <http://trnadb.bioinf.uni-leipzig.de/>), or in two recent studies used LC-MS/MS analysis (Nagayoshi et al. 2021; Li et al. 2020). Table 2 also shows those modifications in other organisms including yeast, mouse and Drosophila. We were not able to confirm Nm residues previously reported in tRNASec_TCA (Nm34) and tRNAVal_AAC(Cm32), however, due to sequence similarity, tRNAVal_AAC clusters together with two other tRNAVal (CAC and TAC1). tRNALeu_AAG and Leu_TAG have similar sequences and thus were not distinguished by sequencing, however Nm32 was detected.

b. Table 2 should also have references for the previous MS studies in humans, Drosophila, and *S. cerevisiae*. In addition, the *S. cerevisiae* data is listed in Table 2 as from MS, but it is also from HPLC without MS.

c. Note also that *S. cerevisiae* tRNA^{Leu}(UAA) has ncm5Um at U34 and not Cm34 as stated in Table 2.

We apologize if these two points were not sufficiently clear. We now make the appropriate changes in the new Table 2 including referencing, adding Mouse in addition to *Drosophila*, Human and *S. cerevisiae*, as well as corresponding referencing in the core text.

2. The data in Fig. 2 and accompanying text should have P-values, so one can judge the importance of the 3.5-fold or greater enrichment in GO categories related to mitochondrial biology.

We apologize if this point was not sufficiently clear. The information was provided in p14 for the GO category discussed in the manuscript: Brain morphogenesis, (FE =7.9 with p-value =7.44E-06 and FDR =4.40E-03). However, for the other GO category, the Reviewer is right, the information was missing in this figure. This information is now included in the corresponding new Figure 2.

3. This reviewer does not understand the data and colors as presented in Figure 3 for the analysis of miRNAs. As it stands it is not clear what the p-values are for any particular comparison or what the colors represent in each column and row of data points from the 9 LCLs examined, and it is not clear to this reviewer what is being sorted. It would help this reviewer if this data were simplified to contain two columns: a single column listing the relative enrichment or downregulation observed in all the LCLs from NSXLID patients (as an

aggregate), relative to that in all the controls (as an aggregate); and another column for p-values for that relative value. Then, individual data from each LCL and miRNA could be shown in the Supplemental figures, with adequate explanation.

We understand the comment of this Reviewer. Indeed the requested file was originally deposited with the raw data at NCBI's Gene Expression Omnibus accessible through GEO Series accession number GSE179384 as well as summarized in Table S1. We believe that the Figure 3A as originally depicted is very informative as it shows not only the log2 FC and p-value for aggregate results (of different control and FTSJ1 mutant cell lines) but also details the difference and variance between the replicates and cell lines as both factors play a role in the p-value. The miRNAs displayed in the heatmap Figure 3A are the first 50 miRNAs deregulated according to the p-value. The purpose of this heatmap is to allow the reader to visualize the detail, such as how each miRNA behaves in the 2 conditions (control and FTSJ1 mutated cells) as well as between cell line replicates and different mutation origin. As said, we understand that one can prefer the aggregate results, for this reason we will make clearer that the corresponding excel file is available at GEO Series accession number GSE179384 in addition to the raw data as well as presented in Table S1 : *FTSJ1 loss of function leads to miRNAs deregulation in NSXLID affected individuals LCLs* with the legend: *A list of the significantly deregulated miRNAs and their log2 fold change and adjusted p-value between FTSJ1 loss-of-function LCLs and control LCLs*. In addition, we make the actual Figure 3 legend clearer (below):

Figure 3. *FTSJ1 loss of function leads to miRNAs deregulation in NSXLID affected individuals LCLs.* (A) Heat map generated using the pheatmap package in R showing the 50 best deregulated miRNAs in p-values, and sorted fold change from most down-regulated (blue) to most up-regulated (red) are represented in two experimental conditions: *FTSJ1 loss of function LCLs (blue turquoise) compared to controls LCLs (pink)*. Condition points to the *FTSJ1 LCL status, WT (Control) or mutated for FTSJ1 gene (FTSJ1)*. The data come from normalized and variance stabilizing transformed read counts using the DESeq2 package in R.

4. The possible connection between miRNA regulation (miR-181a-5p) and BTBD3 could be explored further to determine if it is real.

We have performed KD or OE experiments of *miR-181a-5p* as suggested. We work in HeLa cells for the same reason as explained above, LCL are hardly if not transfectable and both HeLa and LCL similarly express *miR-181a-5p* and *BTBD3*. We included these new experiments performed in HeLa cells in a new supplemental Figure (Figure S4) and corresponding results section as well discuss these new results in the discussion section as depicted below:

Results section:

This cross-analysis revealed that the BTBD3 gene is potentially targeted by miR-181a-5p (He et al, 2015), the two of which were upregulated in NSXLID affected individuals-derived LCLs (Figures 3A, 3C and Table 4), implicating a possible connection between them that differs from the canonical miRNA silencing pathway. LCL are known to be hardly transfectable, however miR-181a-5p and BTBD3 are expressed similarly in HeLa cells (Figure S4A). Thus, by mimicking miR-181a-5p expression or repression, we show that

miR-181a-5p silences BTBD3 in HeLa cells (Figure S4B), strongly suggesting that BTBD3 mRNA is a bona fide target of miR-181a-5p. Strikingly, in FTSJ1 mutant cells, the silencing activity of miR-181a-5p on BTBD3 is compromised in both HeLa and LCL.

Discussion section:

In addition to the reported prediction (He et al. 2015), we show that BTBD3 is a bona fide miR-181a-5p target. Surprisingly, both BTBD3 and miR-181a-5p were up-regulated in FTSJ1 depleted patient cells. While the precise mechanism is not known yet our results suggest that Nm-MTases genes could act upstream of small RNA biogenesis and function through transcriptional downregulation of Argonaute mRNA in Drosophila FTSJ1 mutants (Angelova and Dimitrova et al. 2020) and in human cells (not shown). On the other hand, tRNA fragments (tRF) abundance seen in FTSJ1 mutant fly (Angelova and Dimitrova et al. 2020) and mice (Nagayoshi et al. 2021) can associate with Dicer, Argonaute and Piwi proteins, thus affecting their silencing function. Such tRF-mediated titration of proteins away from canonical substrates has been previously reported in Drosophila and human cell lines (Durdevic et al. 2013; Goodarzi et al. 2015).

5. One major worry about the data in Figure 4 is one is relying entirely on the drug DAP as a proxy for down-regulation of FTSJ, but it would help a lot if this data were accompanied by an FTSJ1 knockdown experiment or equivalent downregulation. That said, the accompanying findings in Drosophila in Fig. 5 are intriguing and more convincing.

We understand the point raised by the Reviewer. However we used this drug because making a KO or KD in human NPC is quite challenging. Importantly the drug was recently validated as a good proxy of FTSJ1 KD by direct binding of DAP to the FTSJ1 enzyme (see Trzaska, C., Amand, S., Bailly, C., Leroy, C., Marchand, V., Duvernois-Berthet, E., Saliou, J.-M., Benhabiles, H., Werkmeister, E., Chassat, T., et al. (2020). 2,6-Diaminopurine as a highly potent corrector of UGA nonsense mutations. Nat. Commun. 11, 1509). In addition and as mentioned by this Reviewer, we show that the effect of DAP mimics the effect of the *Ftsj1* double mutant in *Drosophila*, which is in light with the previous work demonstrating the accuracy of the drug to inactivate FTSJ1.

6. The exciting finding of a long-term memory defect in Drosophila *trm7_34* mutants would benefit greatly from examination of cells lacking *trm7_32* and the corresponding double mutant.

This proposed experiment, *Trm7_32* or/ and *Trm7_32* and *Trm7_34* double mutant on appetitive conditioning assay was performed as requested by the three reviewers. It is now depicted in a new Figure 6 and clearly shows that both positions on tRNA are important for LTM (and not STM). We discuss these new results in the discussion section as depicted below:

A synaptic overgrowth was also observed in Drosophila, indicating that this function of FTSJ1 is conserved across evolution. In addition we found that the long term memory but not the short term was significantly altered in the absence of FTSJ1 in flies. This is consistent with the learning deficits observed in mice and humans. In contrast to Human FTSJ1 and the yeast ortholog TRM7, Drosophila uses two distinct paralogs to methylate positions 32 and 34, respectively, on tRNAs ACL. Interestingly, we found that the lack of

both, Trm7_34 and Trm7_32 had an effect on long term memory, suggesting that the methylation at wobble position 34 and 32 are critical for this function. However, the lack of both modifications (as in mammals Ftsj1 mutant) is not cumulative regarding the memory deficit (Figure. 6). This last observation is strongly supported by the affected human individual that harbours a missense variant (p.Ala26Pro, LCL22 in this study), resulting in loss of Gm₃₄, but not of Cm₃₂ in human tRNA^{Phe} (Guy et al, 2015). Further studies should aim to understand how the loss of methylation at these ACL positions affects the learning and memory functions.

Reviewer #3 (Significance (Required)):

The most important part of the manuscript to this reviewer is the connection between FTSJ1 and neural biology, which would be a major contribution. This connection is begun in Fig. 4-6, but in the opinion of this reviewer, needs more documentation. The beginning of the manuscript is also interesting, but maybe should be part of a separate publication.

We thank the Reviewer 3 for finding our work of interest and suggesting that it could be a major contribution to the field. We understand his/her opinion regarding the first part that is more descriptive. Nevertheless we believe that providing small RNAseq, RNAseq as well as RiboMethSeq Nm detection on 5 different patients' LCLs (including one new mutation described in this manuscript) could be of benefit to a large audience of readers (including human geneticists). Furthermore, to our knowledge, the complete transcriptomics analysis provided here is the first one performed on patients derived cell lines. We believe that the new requested experiments and adding the useful suggestions of Reviewer3 will make the conclusions of our work stronger.

January 4, 2023

RE: Life Science Alliance Manuscript #LSA-2022-01877

Prof. Clément JR Carré
Sorbonne University, Centre National de la Recherche Scientifique, Laboratoire de Biologie du Développement - Institut de Biologie Paris Seine
9 quai Saint Bernard
case courrier 24
Paris, Ile de France 75005
France

Dear Dr. Carré,

Thank you for submitting your revised manuscript entitled "The ribose methylation enzyme FTSJ1 has a conserved role in neuron morphology and learning performance". We would be happy to publish your paper in Life Science Alliance pending final revisions necessary to meet our formatting guidelines.

- please address Reviewer 2's remaining comments
- please upload your main manuscript text as an editable doc file
- please consult our manuscript preparation guidelines <https://www.life-science-alliance.org/manuscript-prep> and make sure your manuscript sections are in the correct order
- please upload all figure files as individual ones, including the supplementary figure files; all figure legends should only appear in the main manuscript file
- please add a Running Title to our system
- please upload your Supplementary Table in editable .doc or excel format
- please add the Twitter handle of your host institute/organization as well as your own or/and one of the authors in our system
- please add your main, supplementary figure, and table legends to the main manuscript text after the references section
- please add an Author Contributions section to your main manuscript text and in the system
- please add a Summary Blurb/Alternate Abstract to our system
- please add a Category for your manuscript in our system
- please upload your Tables in editable .doc or excel format. They can be included at the bottom of the main manuscript file or sent as separate files.
- please remove the DESCRIPTION OF SUPPLEMENTAL DATA section
- please move the DATA AND CODE AVAILABILITY section to the end of the Materials and Methods section, and update so that the deposited datasets are publicly accessible at this point

FIGURE CHECKS:

- please add scale bars to Figure 4B, and indicate their size in the legend

A. FINAL FILES:

- An editable version of the final text (.DOC or .DOCX) is needed for copyediting (no PDFs).

B. MANUSCRIPT ORGANIZATION AND FORMATTING:

Sincerely,

Reviewer #1 (Comments to the Authors (Required)):

The authors have adequately addressed my comments and this exciting paper can now be published.

Reviewer #2 (Comments to the Authors (Required)):

The current work from Carré's group made a comprehensive investigation of tRNA substrates of human FTSJ1 using Ribometh-seq which was performed in patient-derived cells. They further measured the expression profile of mRNA in those cells and observed that deregulated mRNAs are enriched in brain morphogenesis. Interestingly, some specific miRNAs but not general miRNA production were found to be affected, and the FTSJ1 mutation was found to perturb the silencing activity of miR-181a-5p miRNA. Furthermore, they showed that the loss function of FTSJ1 affects human neuronal morphology during development using human Neural Progenitor Cells, and the phenomenon is conserved in Drosophila and is associated with long term memory deficit. Their data strongly supports their conclusion, and connects the function of FTSJ1 with neuronal morphology, which is consistent with the defects of FTSJ1 associated with Non-Syndromic X-Linked Intellectual Disability.

Before acceptance, the manuscript need to be carefully edited in language and format, for example, "stop" or "comma" are missed in some sentence (e.g., P18 line 18) and the references need to be checked, e.g., Guy et al 2021a and Guy et al 2021b are exactly the same paper, and Li et al 2021a and Li et al 2021b are the same paper.

January 10, 2023

RE: Life Science Alliance Manuscript #LSA-2022-01877R

Prof. Clément Carré
Sorbonne University, Centre National de la Recherche Scientifique, Laboratoire de Biologie du Développement - Institut de Biologie Paris Seine
Transgenerational Epigenetics & small RNA Biology (TErBio)
9 quai Saint Bernard
case courrier 24
Paris, Ile de France 75005
France

Dear Dr. Carré,

Thank you for submitting your Research Article entitled "The ribose methylation enzyme FTSJ1 has a conserved role in neuron morphology & learning performance". It is a pleasure to let you know that your manuscript is now accepted for publication in Life Science Alliance. Congratulations on this interesting work.

DISTRIBUTION OF MATERIALS:

Again, congratulations on a very nice paper. I hope you found the review process to be constructive and are pleased with how the manuscript was handled editorially. We look forward to future exciting submissions from your lab.

Sincerely,
